

# The QuantiSlakeTest, dynamic weighting of soil under water to measure soil structural stability

Frédéric M. Vanwindekens[1, *] and Brieuc F. Hardy[1, *]

[1]Department of Sustainability, Systems & Prospective – Unit of Soil, Water and Integrated Crop Production, Walloon Agricultural Research Centre, Rue du Bordia, 4, B-5030 Gembloux, Belgium
[*]These authors contributed equally to this work.

**Correspondence:** Frédéric M. Vanwindekens (f.vanwindekens@cra.wallonie.be) and Brieuc F. Hardy (b.hardy@cra.wallonie.be)

**Abstract.** We evaluated the performance of a new, simple test to evaluate soil structural stability. The QuantiSlakeTest (QST) consists in a quantitative approach of the slake test, a dynamic weighting of a dried structured soil sample once immersed in water. The objective of this work was threefold: we aimed to (i) derive indicators from QST curves to evaluate soil structural stability regarding the underlying mechanisms of soil disaggregation; (ii) establish the relationship between soil properties and

QST indicators; and (iii) assess how QST indicators respond to contrasting soil management practices. To meet these goals, we sampled the soil of 35 plots from three long-term field trials in the silt loam region of Belgium dealing with contrasting organic matter inputs, tillage treatments and P-K fertilisation, respectively. For each plot, QST curves were compared to the three tests of Le Bissonnais, targeting specific mechanisms of soil disaggregation.

Shortly after immersion in water, soil mass increases due to the rapid replacement of air by water in soil porosity. Then

soil mass reaches a maximum before decreasing, once mass loss by disaggregation exceeds mass gain by air loss. Our results confirmed that the early mass loss under water is mainly related to slaking, whereas after a longer time period, clay dispersion becomes the dominant process of soil disaggregation. The overall soil structural stability was positively correlated to the soil organic carbon (SOC) content and negatively correlated to the clay content of soil. Accordingly, the SOC:clay ratio was closely related to QST indicators. Nevertheless, for a similar carbon (C) input, green manure and crop residues were more efficient

in decreasing clay dispersivity whereas farmyard manure promoted SOC storage and was more efficient against slaking. QST curves had a strong discriminating power between reduced tillage and ploughing regardless of the indicator, as reduced tillage increases both total SOC content and root biomass in the topsoil.

The QST has several advantages. It is rapid to run, doesn't require expensive equipment or consumables and provides a high density of information on both specific mechanisms of soil disaggregation and the overall soil structural stability. As an open

access program for QST data management is currently under development, the test has a strong potential for adoption by a widespread community of end-users.



## 1 Introduction

Soil structure is one of the main factors controlling the fertility of temperate agricultural soils subject to intensive cultivation. This is particularly true for Luvisols of the loess belt of Belgium, which are among the most productive soils of Europe and
therefore have experienced a long cropping history. The high productivity of these soils is primarily related to their high plant available water storage capacity as they are deep, free of rocks and with a texture largely dominated by silt, up to 85 % in topsoil. In addition, these soils developed on Quaternary loess deposited $< 170.000$ years ago (Antoine et al., 2003) still contain unweathered primary minerals in the subsoil, acting as a source of nutrients for plants (Vancampenhout et al., 2013). Their clay fraction is dominated by high activity clays, which provides a favourable cation exchange capacity for plant-available
nutrient retention.

Since deforestation centuries ago, the chemical and biological fertility of these soils has increased over the course of cultivation, with topsoil pH, base saturation and earthworm activity increasing under repeated organic and mineral fertiliser application (Langohr, 2001). Nevertheless, today many of these soils have a poor structural stability, which makes them particularly sensitive to physical damages such as compaction and erosion (Bielders et al., 2003). This structural weakness is related to a
silt-dominated texture and exaggerated by a low soil organic matter (SOM) concentration in topsoil. Between 1960s and 2005, cropland soils of the loess belt of Belgium have lost $14\,\mathrm{t\,C\,ha^{-1}}$ on average, mainly caused by a shift from mixed crop-livestock farming systems towards arable farming systems, with a progressive disconnection from animal husbandry. This shift caused a decrease of farmyard manure application on cropland soil and a replacement of cereal and temporary grasslands by spring crops such as sugar beet, potato and chicory (Goidts and van Wesemael, 2007), decreasing soil organic carbon (SOC) inputs. In
parallel, the overall increase in ploughing depth, diluting SOM vertically, has accentuated the decrease in SOC content in the topsoil layer (Meersmans et al., 2009). The Ap horizon of these soils has a typical SOC content of about $10\,\mathrm{g\,kg^{-1}}$ (Meersmans et al., 2011), which is clearly below the threshold value of $12\,\mathrm{g\,kg^{-1}}$ generally considered as critical for structural stability. The combination of a poor soil structural stability with an incomplete soil cover in the winter and spring periods (given the high proportion of spring crops in the rotation) increases erosion risks (Bielders et al., 2003), particularly under the growing
risk of occurrence of extreme climatic events induced by climate change (IPCC, 2014).

In this agricultural context, conservation tillage appears as an effective way to decrease soil susceptibility to erosion and therefore has been increasingly adopted by farmers within the last 20 years. According to local farmers, the replacement of moldboard ploughing by reduced tillage operations such as stubble cultivation seems to have a positive effect on soil structure and water infiltration (Chabert and Sarthou, 2020; Holland, 2004), dramatically decreasing erosion risks (Seitz et al., 2019).
Soil erosion is governed by both rainfall characteristics and environmental factors such as slope characteristics, soil cover and soil properties such as hydraulic conductivity and aggregate stability (Alewell et al., 2019; Maugnard et al., 2013). Owing to the difficulty to measure soil erosion and runoff, soil aggregate stability is often used as an indicator of soil erodibility (Barthès and Roose, 2002). The theory of aggregate hierarchy of Hadas (1987) is widely accepted to conceptualise the internal organisation of soil aggregates. At the lowest level, elementary clay plates ($< 2\,\mu\mathrm{m}$) combine into floccule or domains of clays,
with a degree of organisation depending on clay mineralogy (quasi-crystals > domains > assemblage, Dexter, 1988). Floccule





and domains combine into clusters ($2-20\,\mu m$) under the action of binding agents such as polyvalent cations ($Al^{3+}$ in acidic soils and $Ca^{2+}$ in neutral to slightly basic soils), oxides and organic compounds, mainly polysaccharides from bacterial and fungal mucilages or root exsudates (Dexter, 1988; Tisdall and Oades, 1982). They can be very stable and contain organic acids or partially degraded bio-materials. Clusters combine into micro-aggregates $20-250\,\mu m$ in size (Edwards and Bremner, 1967; Six et al., 2004) that combine themselves into macro-aggregates ($> 250\,\mu m$) under the action of wetting and drying cycles (Dexter, 1988). Roots and fungal hyphae enmeshing micro-aggregates are recognised as critical binding agents in macro-aggregates, and are therefore influenced by soil management practices such as crop rotation and tillage (Tisdall and Oades, 1982). Clods ($> 25\,mm$) constitute the upper level of soil aggregation and are, in many agricultural soils, the result of compaction by agricultural machinery (Dexter, 1988). Under disaggregating forces, it is important to note that the destruction of one hierarchical order automatically destroys all higher hierarchical orders (Dexter, 1988).

Aggregate breakdown is controlled by four mechanisms (Le Bissonnais, 1996; Le Bissonnais and Le Souder, 1995): (i) Slaking occurs during fast-wetting of a soil and consists in the fragmentation of macro-aggregates into micro-aggregates by internal pressure exerted by air entrapment in soil porosity. (ii) Mechanical breakdown by raindrop impact, also known as splash erosion, initiates soil sealing and crusting by liberating elementary particles from soil aggregates. Its amplitude relies on raindrop characteristic as well as internal soil cohesion, which decreases logarithmically with increasing water content (Dexter, 1988). (iii) The breakdown by differential swelling depends on the abundance and swelling properties of clay particles in soil. Nevertheless, this process mainly plays a role at macroscopic scale and has therefore a limited effect on soil disaggregation relative to the other mechanisms (Le Bissonnais, 1996). (iv) Physico-chemical or clay dispersion is the last mechanism, occurring when soil is wet. This dispersion depends on the ionic status of the soil (ionic strength in soil solution and the exchangeable sodium percentage) as well as the mineralogy of clays. Clay dispersion jeopardises the smallest level of soil aggregation (namely quasi-crystals, domains or assemblages of clay particles) to liberate elementary particles, which deteriorates any upper level of soil aggregation (Dexter, 1988).

A large number of methods exist for the measurement of soil structural stability. Methods can be categorised between laboratory and field and methods. Among laboratory methods, most traditional methods are destructive and rely on the resistance of soil aggregates to fragmentation under wet, or, less often, dry conditions. Some wet fragmentation methods rely on the disaggregating power of the wetting treatment only, such as percolation stability (e.g., Mbagwu and Auerswald, 1999; Wuddivira et al., 2009), high Energy Moisture Content (e.g., Levy and Mamedov, 2002), or fast and slow wetting (e.g., Le Bissonnais, 1996). Other methods in wet conditions rely on an additional energy input, such as wet sieving methods (e.g., Hénin et al., 1958; Kemper and Rosenau, 1986; Yoder, 1936), those involving shaking or ultrasonication for clay dispersion (e.g., Haynes, 1993; Zhu et al., 2016), disaggregation by raindrop impact (e.g., Imeson and Vis, 1984) or rainfall simulators (e.g., Loch, 1989). More recently, promising results were obtained with non-destructive methods, such as aggregate delineation by analysis of X-ray microtomography images (Koestel et al., 2021), or aggregate stability prediction by visible-near infrared (VIS-NIR) spectroscopy (Shi et al., 2020). Recently, the SLAKES mobile application provided encouraging results as a tool for rapid data acquisition on soil structure in field conditions. The test relies on image recognition to measure the increase in area of a soil aggregate as it disperses in water (Bagnall and Morgan, 2021; Fajardo et al., 2016; Jones et al., 2021). Among field





methods, a variety of visual soil assessment methods exist, such as the *profil cultural* (Hénin et al., 1960; Gautronneau, 1987), the Peerlkamp field test (Ball et al., 2007), the Visual Evaluation of Soil Structure method (VESS, Guimarães et al., 2017) or the visual soil assessment method (Shepherd et al., 2008). A classification of aggregates according to their stability in water proposed by Emerson (1967) is another approach that can be implemented directly on the field.

The multiplicity of methods of measurement of soil structural stability highlights the complexity of evaluation of soil structure. The preferred approach is a matter of compromise depending on (i) the targeted goal (evaluation of soil structure, management of erosion or compaction risks), (ii) local conditions of soil, topography and climate, (iii) the technicality, cost and delay of measurement; and (iv) the spatial scale of the soil unit to investigate.

In this work, we evaluated the performance of a new, simple test to evaluate soil structural stability, named QuantiSlakeTest
(QST). We propose a quantitative approach of the slake test (QuantiSlakeTest, QST). It consists in the dynamic weighting of a structured soil sample once immersed in demineralised water, in a 8 mm mesh basket. This approach has the advantage to be simple, rapid and dynamic, therefore providing a high density of information all over the process of soil wetting and disaggregation under water.

The objective of this work was threefold: we aimed to (i) unravel the mechanisms controlling soil sample mass evolution
under water and derive indicators from the QST curves to evaluate soil structural stability regarding to related mechanisms of soil disaggregation; (ii) relate QST indicators to soil properties, particularly SOC and clay contents; and (iii) assess how contrasting soil management practices influence the QST indicators.

To meet these goals, we sampled the soil of 35 plots from three long-term field trials of the Walloon agricultural research center (*Centre wallon de recherches agronomiques*, CRA-W) dealing with contrasting farming practices in terms of tillage,
organic matter (OM) restitution and P-K fertilisation. For each plot, we compared the QST indicators to the mean weight diameters (MWD) and the percentage of macro-aggregates (MA, > 200 µm) from the three tests of Le Bissonnais (1996), used as a reference method. Prior to measurements, we were expecting to observe (i) a relative increase in soil structural stability for the soils under reduced tillage compared to ploughing; (ii) a decrease of soil structural stability under long-term K over-fertilisation with KCl (Paradelo et al., 2016) in the P-K mineral fertiliser trial; and (iii) an overall positive correlation
between SOC content and soil structural stability across the dataset (Chan and Mullins, 1994; Fukumasu et al., 2022; Kong et al., 2005; Regelink et al., 2015; Six et al., 2004; Tisdall and Oades, 1982).

## 2 Material and methods

### 2.1 Description of the field trials

The plots sampled for soil structural stability measurements include contrasting treatments from three long-term field trials
dealing with soil tillage, organic matter inputs and P-K mineral fertilisation. At the time of sampling in April 2019, the three trials were covered with winter wheat (*Triticum aestivum*). All field trials are located on the agricultural domain of the CRA-W in Gembloux, a town in the centre of the silt loam region of Wallonia, southern Belgium. The climate is oceanic temperate,



with a mean annual temperature of $10.2°$C and a mean annual rainfall of $793.4\,\text{mm}$ for the 1991-2020 period[1]. All soils are developed from loess, a silt-dominated unconsolidated and free of rock Quaternary sediment (Antoine et al., 2003). Soils are

classified as hortic Luvisols according to the WRB (IUSS Working Group WRB, 2014).

### 2.1.1   Organic matter trial

The organic matter trial (OM trial, $50.560°$ N, $4.726°$ E) was set up in 1959, with the initial goal of addressing the issue of decreasing organic matter inputs (farmyard manure, crop by-products) on cropland soils of the silt loam region and related consequences on soil properties, crop yields and farm profitability (Roisin, 2018). The trial includes six contrasting treatments

of SOM restitution in plots of $70\,\text{m} \times 10\,\text{m}$, repeated six times, following a Latin square design with the blocks aligned in a row. From 1959 to 1974, the field was cropped according to a four-year rotation with sugar beet as the starter crop, followed by three years of winter cereals (wheat, oat, barley) or two winter cereals (wheat, barley/oat) and one legume (horsebean). Cultivation cycle shifted from 1975 onwards to a three-year rotation sugar beet – winter wheat – winter barley. Among the six treatments, three were selected for soil sampling, described by Buysse et al. (2013b). The 'residue exportation' (RE) treatment

consists in a maximal exportation of by-products (straws and sugar beet heads and leaves) and no farmyard manure application nor green manure during the intercropping period. Since 2009 however, sugar beet heads and leaves are left on the field. The 'farmyard manure' (FYM) treatment consists in one application of 30 to $60\,\text{tons}\,\text{ha}^{-1}$ of composted cattle manure once per rotation, after the harvest of the last winter cereal of the rotation in order to enrich the soil for the sugar beet. The last application before soil sampling occurred on the 26th of July 2017. In the 'residue restitution' (RR) treatment, all crop by-products (cereal

straws and sugar beet heads and leaves) are left on the fields, and one cover crop acting as a green manure is sowed once per rotation during the intercropping period between the winter barley and the sugar beet. Cover crops were vetches until 2009 (except once mustard in 1980), phacelia in 2011 and 2014 and a oat-vetch-clover mix in 2017. The annual total carbon (C) input amounts respectively $315 \pm 76\,\text{gC}\,\text{m}^{-2}$, $472 \pm 82\,\text{gC}\,\text{m}^{-2}$ and $487 \pm 93\,\text{gC}\,\text{m}^{-2}$ for the RE, FYM and RR treatments (Buysse et al., 2013a). Since the start of the trial, yearly measurements of topsoil properties ($0-25\,\text{cm}$) show a drop of SOC

content for the RE treatment, an increase for the FYM treatment and a maintain for the RR treatment (Buysse et al., 2013b). For all treatments, soil is ploughed annually with a moldboard plough.

### 2.1.2   Tillage trial

The soil tillage trial ($50.560°$ N, $4.727°$ E) was set up in 2004 and follows a two-year rotation with winter wheat (*Triticum aestivum*) followed by a spring crop, generally sugar beet (*Beta vulgaris*) or flax (*Linum usitatissimum*), alternately, with an

exception in 2018 (*Zea mays L.*). A green manure is sowed after tillage following the harvest of the cereal and destroyed during winter time before the spring crop. Among the four treatments of this trials, the two most contrasting ones were sampled: (i) annual ploughing (P) at a depth of $25-30\,\text{cm}$ with a moldboard plough; (ii) annual reduced tillage (RT) with a spring tine cultivator tilling at a depth of about $10\,\text{cm}$. The two treatments are repeated four times in a complete random block of split-plot type. The plots are $12\,\text{m}$ wide and $21.5\,\text{m}$ long.

---

[1]https://www.meteo.be/resources/climatology/climateCity/pdf/climate_INS92142_9120_fr.pdf





### 2.1.3 P-K mineral fertiliser trial

The P-K mineral fertiliser trial ($50.582°$ N, $4.687°$ E) was set up in 1967, with the initial goal of assessing the effect of the rate of P and K mineral fertiliser application on crop quality and yield, nutrient exportation with harvest, soil properties and farm profitability (Roisin, 2019). The trial comprises three levels of phosphorus (P) fertiliser (applied as superphosphate $18\%$ or triple superphosphate $45\%$) crossed with three levels of potassium (K) fertiliser (applied as KCl 40 or $60\%$), namely nine different treatments repeated six times, for a total of 54 plots of $7.5\,\mathrm{m} \times 50\,\mathrm{m}$. The lower level of P and K fertilisation received no P and K mineral fertiliser since 1975 (P0 and K0). The intermediate level of fertilisation consists in balancing P and K exports by application of the same amount of nutrients through mineral fertilisers, according to the nutrient balance method (P1 and K1). The higher level of fertilisation is over-fertilised, multiplying by 2 (until 2000) or 1.5 (onwards) the amount of P an K applied to the P1 and K1 treatments (P2 and K2). The last application of P-K fertilisers before soil sampling occurred on the 15 July 2016. The whole field is cropped according to a three-year rotation cycle similar to that of the organic matter trial, with sugar beet as starter crop followed by two winter cereals (winter wheat and winter barley). Since the start of the trial, the soil has not received any exogenous organic matter but all by-products (cereal straws and sugar beet heads and leaves) are left on the field to maintain sufficient SOC contents. For all treatments, soil is ploughed annually with a classic moldboard plough, except before the seeding of sugar beet in 2017 (soil was prepared by deep decompaction with a heavy tine cultivator at about $30\,\mathrm{cm}$ depth in august 2016). In this study, we put the focus on the potential effect of contrasting levels of KCl application on soil structural stability, as chlorides are known to weaken soil structure (Paradelo et al., 2016). Therefore, we selected three repetitions of each level of K within the trial.

### 2.2 Soil sampling

Soils were sampled on the 8 and 10 April 2019. For each plot previously described, six structured soil samples of $100\,\mathrm{cm^3}$ were taken with steel Kopecky cylinders, in the inter-row, at a depth of $2-7\,\mathrm{cm}$. Soil was sampled in an area of $1\,\mathrm{m^2}$ that was sprayed three weeks earlier with about $32\,\mathrm{ml}$ of $10\,\mathrm{g\,l^{-1}}$ glyphosate, in order to stop plant growth and therefore standardise sampling conditions between plots at the time of sampling. Soils were carefully transported within the cylinders to the laboratory where they were unmoulded. Five samples were air-dried until constant weight during about three months for QST analysis, whereas the last sample was dried at $105°$C and weighted for the determination of bulk density. Additionally to structured soil samples, about $2\,\mathrm{kg}$ of each soil was sampled at the same location and depth and gently crumbled by hand for the measurement of soil structural stability by the Le Bissonnais (1996) method and analysis of physico-chemical soil properties.

### 2.3 Soil analysis

#### 2.3.1 Physico-chemical properties of soils

After homogeneization, about $500\,\mathrm{g}$ of each disturbed soil sample were gently crushed with a pie roll and sieved to $2\,\mathrm{mm}$, and the fraction $< 2\,\mathrm{mm}$ was sent to the *Centre interprovincial de l'agriculture et de la ruralité* in La Hulpe (Belgium)



to be analysed. Soil pH was measured in water ($\text{pH}_{\text{H}_2\text{O}}$) with a $1:5$ soil:solution mass ratio, according to the norm NF-ISO-10390:2005 (International Organization for Standardization, 2005). Total C content was determined by dry combustion according to the norm NF-ISO-10694:1995 (International Organization for Standardization, 1995). Inorganic C content was measured by infrared quantification of $CO_2$ emitted from soil after addition of orthophosphoric acid, according to the norm NF-EN-15936:2012 (Association Française de Normalisation, 2012). SOC content was calculated as the difference between total and inorganic C content. Granulometry analysis (sand, silt and clay contents) was made by sedimentation and sieving, according to Stokes law, by a method derived from the norm NF-X31-107:2003 (Association Française de Normalisation, 2003). Bulk density was measured from one structured soil sample per plot, dried at $105°C$ until constant weight, by dividing the mass of dry soil by the core volume ($100\,\text{cm}^3$). The main properties of the soils of the experimental fields of the three trials are shown in table 1.

### 2.3.2 Measurement of soil structural stability by *Le Bissonnais* method

Soil structural stability was measured according to the method of Le Bissonnais (1996), following the norm ISO-FDIS-10930:2011 (International Organization for Standardization, 2012). For each soil gently crumbled by hand, $5\,\text{g}$ to $10\,\text{g}$ of soil aggregates from 3 to $5\,\text{mm}$ in size were subjected to three contrasting disaggregating treatments. The first test consists in a fast-wetting of soil aggregates in water, to test their resistance to slaking. The second test is a slow-wetting of soil aggregates by capillarity, to test their resistance to clay dispersion and swelling in wet conditions. The third test consists in a standardised shaking of the aggregates in water after rewetting them in $95\,\%$ v/v ethanol for $30\,\text{min}$, to test their mechanical strength besides of the slaking effect. After each disaggregation treatments, the aggregates were immersed in ethanol and dried at $40°C$ for $2\,\text{hours}$. The size distribution of the remaining aggregates was measured by way of dry sieving, with sieves of $2\,\text{mm}$, $1\,\text{mm}$, $0.5\,\text{mm}$, $0.2\,\text{mm}$, $0.1\,\text{mm}$ and $0.05\,\text{mm}$. Two main indicators were calculated from the fractions. The first is the mean weighted diameter (MWD) of the aggregate fraction that survived each individual test, following the equation (International Organization for Standardization, 2012):

$$MWD = \frac{\sum(mean\ diameter\ between\ two\ sieves \times [weighted\ percentage\ of\ particles\ retained\ on\ the\ sieve])}{100} \quad (1)$$

The second is the percentage of macro-aggregates (MA) remaining after each individual test, calculated as the mass fraction of soil aggregates $> 200\,\mu\text{m}$.

The Le Bissonnais method has two main advantages. First, the three tests target the three main mechanisms of soil disaggregation in field conditions, namely slaking, raindrop impact and clay dispersion; and second, it measures the size distribution of particles remaining after the disaggregation treatment, which provides further insight in soil susceptibility to water erosion (Le Bissonnais, 1996).



**Table 1.** Soil properties of the 35 plots from the three long-term field trials. SOC = Soil Organic Carbon. The SOC:clay ratio was calculated for harmonised units for SOC and Clay, $g\,kg^{-1}$.

| Plot | Treatment | Clay | Silt (fine) | Silt (coarse) | Silt (total) | Sand (fine) | Sand (coarse) | Sand (total) | SOC | SOC:clay | pH$_{H_2O}$ | Bulk density |
|---|---|---|---|---|---|---|---|---|---|---|---|---|
| | | % | % | % | % | % | % | % | $g\,kg^{-1}$ | [-] | [-] | $g\,cm^{-3}$ |
| Organic matter trial | | | | | | | | | | | | |
| 1 | Farmyard manure | 16.6 | 30.9 | 46.0 | 76.9 | 5.3 | 1.2 | 6.5 | 13.66 | 0.082 | 7.37 | 1.31 |
| 2 | Farmyard manure | 19.7 | 31.1 | 43.6 | 74.7 | 4.5 | 1.1 | 5.6 | 10.88 | 0.055 | 7.22 | 1.32 |
| 3 | Farmyard manure | 18.6 | 30.0 | 45.5 | 75.5 | 4.9 | 1.0 | 5.9 | 11.16 | 0.060 | 7.16 | 1.32 |
| 4 | Residue exportation | 16.1 | 29.2 | 48.4 | 77.5 | 5.4 | 1.0 | 6.4 | 8.82 | 0.055 | 7.07 | 1.28 |
| 5 | Residue restitution | 15.1 | 29.9 | 49.2 | 79.1 | 4.7 | 1.1 | 5.8 | 9.66 | 0.064 | 6.93 | 1.34 |
| 6 | Residue restitution | 14.8 | 30.7 | 48.1 | 78.8 | 4.8 | 1.6 | 6.4 | 9.84 | 0.067 | 6.86 | 1.30 |
| 7 | Residue exportation | 14.0 | 29.2 | 50.4 | 79.6 | 5.2 | 1.2 | 6.4 | 8.85 | 0.063 | 7.04 | 1.30 |
| 8 | Farmyard manure | 13.7 | 30.1 | 49.6 | 79.7 | 5.2 | 1.4 | 6.6 | 10.59 | 0.077 | 6.83 | 1.30 |
| 9 | Residue exportation | 15.8 | 29.3 | 48.8 | 78.0 | 5.1 | 1.1 | 6.2 | 8.80 | 0.056 | 6.88 | 1.34 |
| 10 | Residue restitution | 15.3 | 30.6 | 48.2 | 78.8 | 4.7 | 1.2 | 5.9 | 11.19 | 0.073 | 6.91 | 1.30 |
| 11 | Residue restitution | 18.9 | 31.0 | 44.6 | 75.6 | 4.7 | 0.8 | 5.6 | 9.84 | 0.052 | 6.99 | 1.32 |
| 12 | Farmyard manure | 15.3 | 29.9 | 48.6 | 78.5 | 4.7 | 1.5 | 6.2 | 10.22 | 0.067 | 6.75 | 1.36 |
| 13 | Residue exportation | 17.3 | 29.8 | 47.1 | 76.9 | 4.6 | 1.2 | 5.7 | 8.02 | 0.046 | 7.14 | 1.27 |
| 14 | Farmyard manure | 14.4 | 29.5 | 49.2 | 78.8 | 5.4 | 1.4 | 6.8 | 11.39 | 0.079 | 7.12 | 1.28 |
| 15 | Residue restitution | 17.1 | 30.1 | 46.8 | 76.9 | 5.0 | 1.0 | 6.0 | 9.81 | 0.057 | 6.98 | 1.30 |
| 16 | Residue exportation | 19.3 | 29.5 | 45.4 | 74.9 | 4.9 | 0.9 | 5.8 | 8.23 | 0.043 | 6.82 | 1.33 |
| 17 | Residue restitution | 19.0 | 31.2 | 44.2 | 75.4 | 4.7 | 0.9 | 5.5 | 9.36 | 0.049 | 7.08 | 1.32 |
| 18 | Residue exportation | 20.0 | 32.5 | 42.3 | 74.8 | 4.3 | 0.8 | 5.2 | 7.76 | 0.039 | 7.21 | 1.37 |
| Tillage trial | | | | | | | | | | | | |
| 19 | Reduced tillage | 13.1 | 30.4 | 49.2 | 79.6 | 5.9 | 1.5 | 7.4 | 12.99 | 0.099 | 7.17 | 1.21 |
| 20 | Ploughing | 16.7 | 28.5 | 48.2 | 76.7 | 5.2 | 1.3 | 6.6 | 9.95 | 0.060 | 7.49 | 1.27 |
| 21 | Reduced tillage | 16.7 | 30.3 | 46.9 | 77.2 | 5.0 | 1.1 | 6.1 | 11.19 | 0.067 | 7.35 | 1.25 |
| 22 | Ploughing | 15.0 | 32.4 | 46.2 | 78.6 | 4.7 | 1.7 | 6.4 | 9.87 | 0.066 | 7.75 | 1.28 |
| 23 | Ploughing | 12.5 | 28.9 | 52.0 | 80.9 | 5.2 | 1.4 | 6.6 | 9.04 | 0.072 | 7.72 | 1.27 |
| 24 | Reduced tillage | 12.6 | 30.6 | 50.0 | 80.6 | 5.5 | 1.3 | 6.8 | 11.99 | 0.095 | 7.50 | 1.21 |
| 25 | Ploughing | 11.7 | 29.2 | 51.5 | 80.6 | 5.9 | 1.7 | 7.6 | 11.44 | 0.097 | 7.59 | 1.26 |
| 26 | Reduced tillage | 11.7 | 29.5 | 52.5 | 82.0 | 5.2 | 1.2 | 6.3 | 12.49 | 0.107 | 6.88 | 1.22 |
| P-K mineral fertiliser trial | | | | | | | | | | | | |
| 27 | K0 | 19.9 | 27.5 | 47.4 | 74.9 | 5.1 | 0.2 | 5.3 | 11.92 | 0.060 | 6.86 | 1.28 |
| 28 | K1 | 20.0 | 28.3 | 46.0 | 74.2 | 5.5 | 0.3 | 5.8 | 9.75 | 0.049 | 6.91 | 1.31 |
| 29 | K2 | 19.9 | 26.8 | 47.4 | 74.2 | 5.6 | 0.4 | 6.0 | 11.40 | 0.057 | 7.00 | 1.30 |
| 30 | K1 | 16.2 | 27.9 | 49.8 | 77.7 | 5.6 | 0.5 | 6.1 | 10.83 | 0.067 | 6.61 | 1.35 |
| 31 | K0 | 15.0 | 28.0 | 50.3 | 78.3 | 6.1 | 0.6 | 6.7 | 11.52 | 0.077 | 6.95 | 1.25 |
| 32 | K2 | 16.6 | 30.3 | 47.3 | 77.5 | 5.2 | 0.7 | 5.8 | 11.74 | 0.071 | 6.80 | 1.39 |
| 33 | K2 | 16.9 | 31.9 | 46.2 | 78.1 | 4.4 | 0.7 | 5.0 | 12.50 | 0.074 | 6.85 | 1.33 |
| 34 | K0 | 12.8 | 28.7 | 52.4 | 81.0 | 5.8 | 0.4 | 6.2 | 11.03 | 0.086 | 6.69 | 1.28 |
| 35 | K1 | 13.0 | 31.3 | 50.3 | 81.6 | 5.1 | 0.3 | 5.4 | 9.69 | 0.074 | 6.69 | 1.27 |



### 2.3.3 Soil structural stability measurement by the QuantiSlake Test (QST) method

The QST method consists in introducing a structured soil sample supported by a $8\,mm$ metallic mesh basket into distilled water, and measuring soil mass continuously by dynamically weighting the content of the basket using the underfloor weighting hook of the balance. The balance is connected to a computer for datalogging (Fig. 1). For each plot, the five air-dried structured soil samples were slaked during approximately $1000\,sec$ (around $17\,min$), with a recording time frequency decreasing over time under water, from less than one second at the start of the experiment to approximately $30\,s$ at the end. Due to some electronic or computer issues during the experiment, some samples were lost. At the end, the data from 157 QST could be processed and constituted the main data base of our study. Most of the 35 plots gave five or four usable QST curves (20 and 13 plots respectively). One plot from the OM trial and one from the P-K mineral fertiliser trial gave only three and two usable curves, respectively.

Immediately after soil immersion in water, soil mass drops due to Archimedes' upward buoyant force. This first value of soil mass under water (right after Archimedes' buoyancy) is defined as the time 0 ($t_0$) of the QST test. Then soil mass increases due to the release of air and the infilling of soil porosity by water. After a few seconds or minutes, the mass of cropland soils reaches a maximum ($W_{max}$) before decreasing due to disaggregation. Soil mass was normalised according to the maximum mass value reached by each individual sample, so that mass values vary between 0 and 1. Several indicators were calculated from the QST curves (Fig. 2). QST indicators were split into four categories :

- (i) indicators related to the early increase in soil mass soon after soil immersion in water; they include the time to reach the maximum mass value ($t_{max}$); the increase in soil mass between $t_0$ and $t_{max}$ ($W_{max}$-$W_{t0}$); and the slope between $t_0$ and $t_{max}$ ($Slope_{t0\text{-}max}$);

- (ii) indicators related to the early to intermediate mass loss after reaching the maximum mass; they consists in slopes at different timesteps (after $30\,s$, $60\,s$, $300\,s$ and $600\,s$) in the decreasing part of the curve, taking $t_{max}$ as the starting point ($Slope_{max\text{-}30}$, $Slope_{max\text{-}60}$, $Slope_{max\text{-}300}$ and $Slope_{max\text{-}600}$);

- (iii) indicators specific to the intermediate to late mass loss of soil. They correspond to the time needed to reach a certain fraction of total mass loss between the maximum and the final mass of soil at the end of the QST experiment. Threshold values of 25, 50, 75, 90 and 95 % of mass loss were calculated (t25, t50, t75, t90 and t95); and

- (iv) global indicators providing a complete overview of soil mass evolution all over the QST. They include sample mass at the end of the experiment ($W_{end}$) and the Area Under Curve (AUC);

Calculation of each QST indicator is illustrated in Fig. 2.
thick square light grey boxes thin round light grey boxes




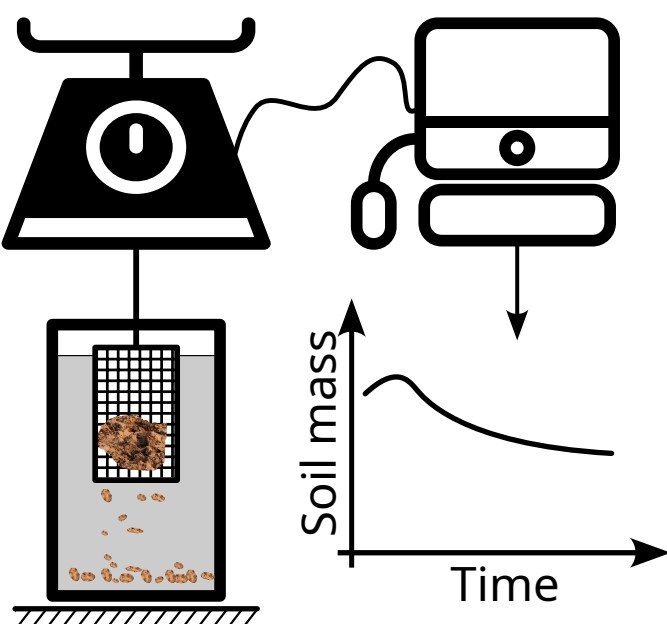

**Figure 1.** Illustration of the QuantiSlake Test device, consisting in the dynamic weighting of a structured soil sample suspended into distilled water by a $8\,\mathrm{mm}$ mesh metallic basket. The balance is connected to a computer for direct datalogging. The construction of an opensource user interface for managing QST laboratory and for visualising data is currently in progress. Video comparing the QST of two contrasting soil samples can be watched here: https://youtu.be/G9UweThvHYI – Illustration based on two graphics by Adnen Kadri from the Noun Project

### 2.3.4 Measurement of root biomass after slaking

For samples from the soil tillage trial, root biomass retained in the metallic basket were weighted after running the QST by cleaning remaining soil with a water jet. The roots were dried carefully with a Tork paper and weighted.

### 2.3.5 Statistical analysis

In order to test if soil management practices affect QST indicators, Linear Mixed-Effects Models were fitted and tested by analysis of variance (ANOVA). For each test, the QST indicator was used as the outcome variable and the treatments of the
trial were used as a fixed explanatory variable, whereas the blocks were defined as a random effect. As several samples were related to one single plot (157 QST in total from 35 plots), the plot identifier was added as a random effect of the model to take into account the dependence between field repetitions from one plot.

Prior to the ANOVA, the normality and the homoscedasticity of the residuals of the models were verified using respectively Shapiro-Wilk and Bartlett tests. For all the models, the significance of differences of QST indicators between soil management
practices were tested using classical analysis of variance (ANOVA, Type II Wald F tests with Kenward-Roger estimation of



**Figure 2.** Illustration of main QST parameters calculation from a curve (slake test on a soil sample from the plot 29, P-K mineral fertiliser trial, see table 1). (i) Inside thick square light grey boxes (upper left), indicators related to the early increase in soil mass including time to reach the maximum mass value ($t_{max}$); the increase in soil mass between $t_0$ and $t_{max}$ ($W_{max}-W_{t0}$); and the slope between $t_0$ and $t_{max}$ ($Slope_{t0-max}$). (ii) In thick round white boxes (upper right), slopes at different timesteps (after 60 s, 300 s and 600 s) in the decreasing part of the curve, taking $t_{max}$ as the starting point ($Slope_{max-60}$, $Slope_{max-300}$ and $Slope_{max-600}$). For the sake of clarity, the value of $Slope_{max-30}$ is not shown in the figure. (iii) In thin round white boxes (below the curve), threshold values of 25, 50, 75, 90,95 and 99% of mass loss (t25, t50, t75, t90, t95 and t99). And finally, (iv) in black boxes, the two global indicators including sample mass at the end of the experiment ($W_{end}$) and the Area Under Curve (AUC, shaded area).





degree of freedom, Fox and Weisberg, 2019). When the F-test was significant ($p < 0.1$), post-hoc comparisons were performed: treatments of the trial were compared pairwise at 0.1 probability level of significance using estimated marginal means (EMMs, also named least-squares means, Lenth, 2022).

Between continuous variables, correlation coefficients were determined. For QST indicators, average values were calculated at the plot level for comparison with other data (structural stability indicators from Le Bissonnais, physico-chemical properties) that were measured only at the plot level.

All statistical analyses were performed using R-4.2.1 software (R Core Team, 2022). The linear mixed-effect models were performed with the lme4 package (Bates et al., 2015), the ANOVA with the car package (Fox and Weisberg, 2019) and contrast analyses with the emmeans package (Lenth, 2022).

## 265 3 Results

### 3.1 Comparison of QST indicators with Le Bissonnais

Except for the $Slope_{t0-max}$, a positive correlation was found between all QST indicators and the mean weight diameter (MWD1) and the percentage of macro-aggregates (MA1) of the fast wetting test of Le Bissonnais (Fig. 2). The higher correlation coefficients were found for QST indicators related to the early stage of the curve, namely $t_{max}$, $W_{max}$-$W_{t0}$, $Slope_{max-30}$, $Slope_{max-60}$,

t25 and t50. Correlation decreases progressively for later slopes ($Slope_{max-300}$, $Slope_{max-600}$) as well as for t75 to t95 and is minimal for sample residual mass at the end of the test ($W_{end}$). Similarly, the mean weight diameters (MWD2) of the slow wetting test of Le Bissonnais also correlate positively with each QST indicator except Slope 0-max. However, correlation tend to increase for QST indicators related to the intermediate to late stage of the curve, particularly t50 to t95 (Fig. 2). In contrast to the fast wetting test, the percentage of macro-aggregates surviving the slow wetting (MA2) are poorly related to QST indi-

cators. For the third test of Le Bissonnais, testing soil mechanical strength, mean weight diameter (MWD3) correlates poorly with QST indicators. Similarly, correlation between QST indicators and the percentage of macro-aggregates surviving the third test (MA3) is always negative and generally poor, except for $W_{max}$-$W_{t0}$ (r=-0.60, Fig. 2). Regardless of the test, sample mass at the end of the experiment ($W_{end}$) correlate poorly with MWDs from Le Bissonnais, considered alone or in combination (data not shown). Correlation between MWD1 (r=0.42) and MWD2 (r=0.38) and the area under curve (AUC) is a bit higher but

remains poor.

### 3.2 Soil structural stability against soil properties

#### 3.2.1 Le Bissonnais

A positive correlation exists between total SOC content and both MWD1 (r=0.75) and MWD2 (r=0.70), whereas MWD3 and MA3 correlate poorly with SOC content (r=0.11 and -0.07, respectively). In contrast, clay content correlates positively

with MWD3 and MA3 (r= 0.52 and 0.66, respectively) but poorly with MWD1 and MWD2 (r=-0.35 and -0.12). Linear relationship with the SOC:clay ratio, evidenced as a proxy for predicting field soil structural stability by visual assessment



| | Le Bissonnais *et al.* (1996) | | | | | | Soil properties | | | | |
| | MWD 1 | MWD 2 | MWD 3 | MA 1 | MA 2 | MA 3 | SOC | Clay | SOC:Clay | pH | Bulk density |
| **QST indicators** | | | | | | | | | | | |
| Slope 0-max | -0.121 | -0.150 | 0.094 | -0.194 | 0.061 | 0.322 | -0.025 | 0.393 | -0.336 | -0.082 | 0.085 |
| tmax | 0.602 | 0.505 | -0.260 | 0.570 | -0.058 | -0.462 | 0.524 | -0.675 | 0.822 | 0.030 | -0.490 |
| Wmax-Wt0 | 0.577 | 0.362 | -0.370 | 0.572 | -0.119 | -0.603 | 0.557 | -0.831 | 0.925 | 0.082 | -0.605 |
| Slope max-30 | 0.465 | 0.372 | -0.160 | 0.485 | 0.022 | -0.357 | 0.409 | -0.638 | 0.672 | -0.031 | -0.142 |
| Slope max-60 | 0.543 | 0.482 | -0.078 | 0.512 | 0.052 | -0.301 | 0.462 | -0.593 | 0.689 | -0.101 | -0.206 |
| Slope max-300 | 0.405 | 0.392 | 0.033 | 0.330 | -0.100 | -0.257 | 0.295 | -0.562 | 0.597 | 0.012 | -0.391 |
| Slope max-600 | 0.355 | 0.309 | 0.033 | 0.293 | -0.125 | -0.260 | 0.243 | -0.569 | 0.570 | 0.031 | -0.430 |
| t25 | 0.602 | 0.498 | -0.178 | 0.570 | 0.039 | -0.324 | 0.518 | -0.510 | 0.681 | -0.084 | -0.179 |
| t50 | 0.573 | 0.556 | -0.133 | 0.519 | 0.069 | -0.281 | 0.499 | -0.426 | 0.633 | -0.053 | -0.212 |
| t75 | 0.487 | 0.665 | -0.065 | 0.430 | 0.152 | -0.213 | 0.512 | -0.331 | 0.584 | 0.007 | -0.291 |
| t90 | 0.428 | 0.594 | -0.078 | 0.382 | 0.163 | -0.254 | 0.477 | -0.342 | 0.582 | 0.046 | -0.313 |
| t95 | 0.387 | 0.639 | -0.091 | 0.357 | 0.224 | -0.273 | 0.552 | -0.351 | 0.610 | 0.148 | -0.343 |
| t99 | 0.169 | 0.292 | -0.259 | 0.184 | 0.017 | -0.247 | 0.313 | -0.302 | 0.384 | 0.185 | -0.161 |
| Wend | 0.328 | 0.275 | 0.033 | 0.261 | -0.171 | -0.246 | 0.196 | -0.542 | 0.526 | -0.001 | -0.391 |
| AUC | 0.421 | 0.379 | -0.011 | 0.351 | -0.103 | -0.295 | 0.300 | -0.593 | 0.618 | 0.022 | -0.401 |

**Table 2.** Correlation coefficients between average QST indicators calculated from individual curves, Mean Weight Diameters (MWD) and percentages of macro-aggregates (MA) from the three tests of Le Bissonnais (1. Fast wetting; 2. Slow wetting; 3. Shaking in water after rewetting with EtOH) and soil properties. The gradient of colours relates to the positive (blues) or to the negative (oranges) relative amplitude of correlation coefficients.





methods (Johannes et al., 2017) was also tested. The SOC:clay ratio correlated positively with both MWD1 (r=0.67) and MWD2 (r=0.48) and negatively with MWD3 (r=-0.33) and MA3 (r=-0.55). No clear linear relationship was found between Le Bissonnais's indicators and pH or bulk density.

### 3.2.2   QuantiSlake test

Except for the $Slope_{0-max}$, indicators derived from QST curves correlate all positively with SOC content. Coefficients remain low to moderate though, with the stronger coefficient obtained for $W_{max}$-$W_{t0}$ (r=0.56) and t95 (r=0.55) (Fig. 2). In contrast, all QST indicators correlate negatively with clay content. The stronger coefficients were found for $W_{max}$-$W_{t0}$ (r=-0.83), $t_{max}$ (r=-0.68), $Slope_{max-30}$ (r=-0.64) and AUC (r=-0.59) (Fig. 2). This seemingly antagonist effect of SOC and clay contents on soil resistance to disaggregation under water is well captured by the SOC:clay ratio, which correlates strongly with indicators from the start of QST curves, particularly $W_{max}$-$W_{t0}$ (r=0.925, Fig. 2 and detailed in Fig. 3) but also $t_{max}$ (r=0.82), $Slope_{max-30}$ (r=0.67), $Slope_{max-60}$ (r=0.69) and t25 (r=0.68). Similarly to indicators of soil structural stability from Le Bissonnais, all indicators from the QST curves correlated poorly with pH. Except for $Slope_{0-max}$, a moderate to poor negative correlation is observed between QST indicators and bulk density, with the lower values obtained for $W_{max}$-$W_{t0}$ (r=-0.61) and $t_{max}$ (r=-0.49).

### 3.3   Soil structural stability under contrasting agricultural practices

The responses of soil structural stability indicators calculated from QST curves to contrasting long-term soil management practices from the three field trials are presented in this section. For the sake of clarity, we put the focus on a selection of nine indicators representative for (i) the start of the curve ($W_{max}$-$W_0$, $t_{max}$, $Slope_{max-30}$, $Slope_{max-60}$), (ii) intermediate and late stages of the curve ($Slope_{max-300}$, t75, t95) and (iii) global QST indicators ($W_{end}$, AUC).

### 3.3.1   Organic matter trial

Soils of the three treatments of OM inputs in the OM trial have different contents of total SOC, with the FYM treatment having the highest SOC content ($11.32\,\mathrm{g\,kg^{-1}}$), the RE treatment having the lowest SOC content ($8.41\,\mathrm{g\,kg^{-1}}$) and the RR treatment having intermediate values ($9.95\,\mathrm{g\,kg^{-1}}$). QST indicators from the start of the QST curves ($W_{max}$-$W_{t0}$, $t_{max}$, $Slope_{max-30}$) tend to respect this gradient of total SOC, with the FYM and RR treatments showing better scores on average than the RE treatment, even if differences are small and only significant ($p < 0.1$) for $t_{max}$ (Fig. 4a-d). Counter-intuitively, this order is not respected anymore for other QST indicators related to intermediate or late stages of the curves. The response of treatments follows the order RR > FYM > RE for $Slope_{max-60}$ (n.s., Fig. 4d) and $Slope_{max-300}$ ($p < 0.1$, Fig. 4e), and FYM shows an average score even lower than RE (n.s.) and significantly lower than RR for $W_{end}$ ($p < 0.1$, Fig. 4h). Conflicting results were also obtained between the three tests of Le Bissonnais, with the MWD scores from the fast wetting test (MWD1) and from the shaking (MWD3) in favor of the FYM treatment (FYM ~ RR > RE) but the scores from the slow wetting test in favor of the RR treatment (RR > FYM ~ RE).





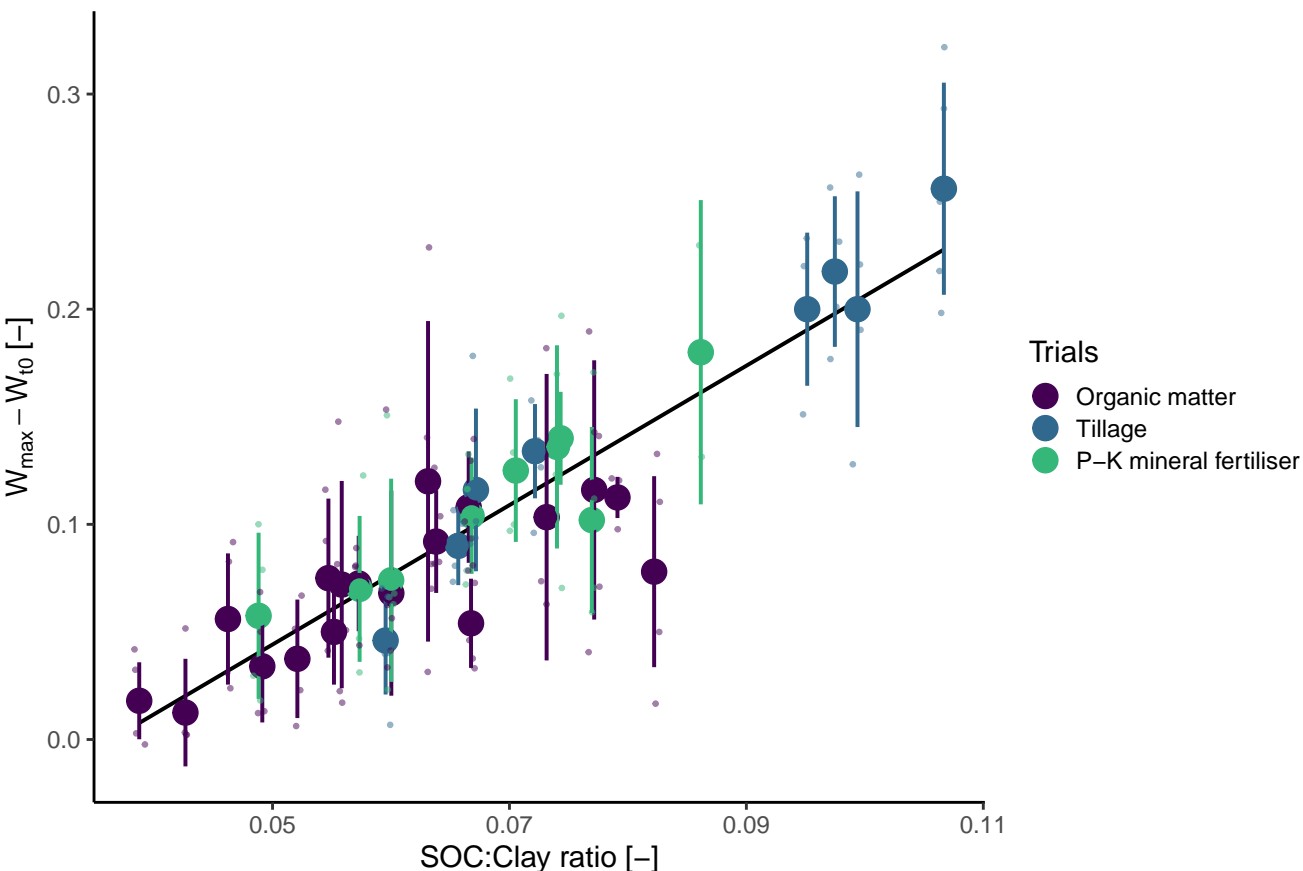

**Figure 3.** Early increase in soil mass under water measured from QST curves ($W_{max}$-$W_{t0}$) against the SOC:clay ratio of bulk soil (r=0.925). Small dots are individual QST indicator with a small amount of noise added in x. Large dots are the mean (M) and bars are standard deviation (sd) by plot (M-sd, M+sd).

### 3.3.2 Tillage trial

Remarkably, the QST responds very well to contrasting tillage treatments, with all QST indicators having a better score for reduced tillage (RT) than for ploughing (P) (Fig. 6). This result is in agreement with total SOC content, RT having an average SOC content of $12.16\,\mathrm{g\,kg^{-1}}$ whereas P treatments have an average SOC content of $10.07\,\mathrm{g\,kg^{-1}}$. However, $Slope_{max-60}$, $Slope_{max-300}$, $Slope_{max-600}$ (respectively $p < 0.05$, $p < 0.01$, $p < 0.01$) and global QST indicators ($W_{end}$, AUC, respectively, $p < 0.01$ and $p < 0.05$, Fig. 6d-e,h-i) tend to discriminate better between tillage treatments than indicators from the start of the curve ($W_{max}$-$W_{t0}$, $t_{max}$ and $Slope_{max-30}$, respectively $p < 0.1$, n.s. and n.s., Fig. 6a-c). Indicators from the three tests of Le Bissonnais provide similar results, with the most contrasting response between RT and P tillage treatments obtained for the fast wetting test. However, $Slope_{max-300}$, $Slope_{max-600}$ and $W_{end}$ discriminate better between tillage treatments than MWD1.



**Figure 4.** Boxplots of nine QST indicators against treatments of OM input for the soils from the organic matter trial, 'residue exportation' (RE), 'farmyard manure' (FYM) and 'residue restitution' (RR). a) $W_{max}$-$W_{t0}$; b) $t_{max}$; c) $Slope_{max-30}$; d) $Slope_{max-60}$; e) $Slope_{max-300}$; f) t75; g) t95; h) $W_{end}$; i) AUC.



**(a)**  **(b)**

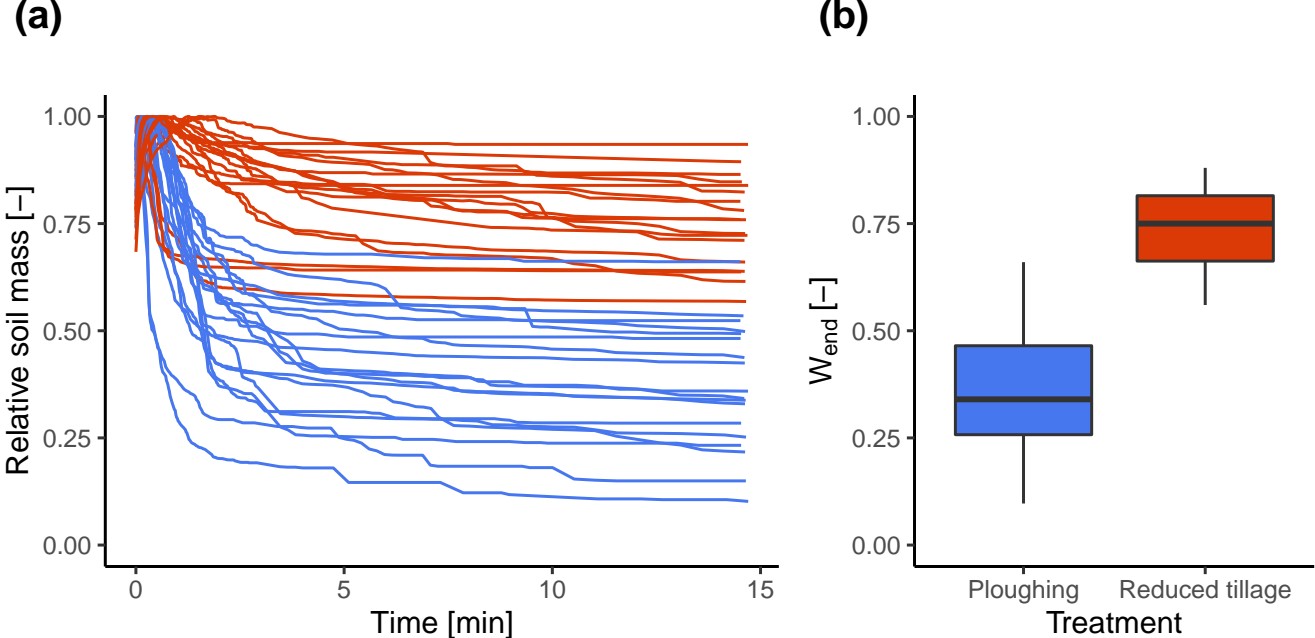

**Figure 5.** QST curves (a) and final mass ($W_{end}$, b) for ploughing and reduced tillage treatments of the tillage trial.

### 3.3.3 P-K mineral fertiliser trial

In the P-K mineral fertiliser trial, soil structural stability respects the order K2 > K0 > K1 regardless of the QST indicator, but without any significant differences (n.s.). Similar results were obtained with the three tests of Le Bissonnais but with a smaller standard deviation on average than that for QST indicators.

## 4 Discussion

### 4.1 Interpretation of QST curves in light of mechanisms of soil disaggregation and soil properties

#### 4.1.1 Mechanisms of soil disaggregation

Right after immersion in water, soil mass increases due to the replacement of air by water in soil porosity. Sooner or later, soil mass then reaches a maximum before decreasing when mass loss by disaggregation exceeds mass gain due to infilling of soil 335 porosity with water. QST indicators from the start of the curves ($W_{max}$-$W_0$, $t_{max}$, Slope$_{max-30}$, Slope$_{max-60}$, t25) are the most correlated to the fast wetting test of Le Bissonnais (MWD1 and MA1; Fig. 2), which indicates that the early mass loss under water is mainly controlled by slaking. In contrast, QST indicators from the intermediate to late stages of the curves (t50 to t95) are more correlated to the slow-wetting test of Le Bissonnais (Fig. 2), specifically targeting clay dispersion. This indicates



**Figure 6.** Boxplots of nine QST indicators against tillage treatments, ploughing (P) and reduced tillage (RT) for the soils from the tillage trial. a) $W_{max}$-$W_{t0}$; b) $t_{max}$; c) $Slope_{max-30}$; d) $Slope_{max-60}$; e) $Slope_{max-300}$; f) t75; g) t95; h) $W_{end}$; i) AUC.





that after a longer time period under water, when soil is saturated, the effect of slaking decreases and clay dispersion becomes

the dominant mechanism of soil disaggregation. Nevertheless, both mechanisms overlap, with air release from soil further interfering with the measurement of soil mass loss when running QST. This may explain for the relatively low correlation coefficients obtained between QST slopes and indicators from the fast and slow wetting tests of Le Bissonnais. We also advocate that the fast-wetting test of Le Bissonnais is unable to measure the effect of slaking independently from the effect of clay dispersion, and rather provides a measurement of the effect of slaking and clay dispersion combined. Indeed, our results

suggest that, for the silt loam soils low in SOC content of this study, the timing of 10 minutes under water recommended for the fast-wetting test of Le Bissonnais is too long to specifically target slaking, since air release from the sample lasted much less than 10 minutes. Accordingly, the final mass ($W_{end}$) of the sample was often reached, or close to, after 10 minutes. Therefore, we think that some QST indicators might be more specific to slaking than the fast-wetting test of Le Bissonnais, such as $Slope_{max-60}$.

Excepted for $W_{max}$-$W_{t0}$, QST indicators are very poorly correlated to the third test of Le Bissonnais, targeting soil resistance to raindrop impact (Le Bissonnais, 1996). This indicates that little information on soil resistance to raindrop impact can be inferred from QST curves, which is not surprising. For the soils of this study, soil resistance to raindrop impact (as estimated by the third test of LeBissonnais) seems to be somehow controlled by the absolute clay content of soil, since clay content correlates positively to MWD3 (r=0.52) and MA3 (r=0.66).

### 4.1.2 The response of QST indicators to soil properties

Soil mass evolution under water as captured by QST indicators respond in an antagonist way to SOC and clay contents. Indeed, all QST indicators are positively correlated to SOC content and negatively correlated to clay content, with the absolute value of correlation coefficients decreasing for indicators of the later part of the curves ($Slope_{max-300}$, $Slope_{max-600}$ and $W_{end}$). Similar trends were observed in other contexts, with the resistance to slaking increasing with SOC content and decreasing with clay

content (Jones et al., 2021; Francis et al., 2019). In light of the comparison between QST curves and Le Bissonnais's indicators, the amplitude of the early mass loss under water is mainly controlled by soil resistance to slaking. Accordingly, the absolute SOC content increases soil resistance to slaking, as highlighted by the positive correlation between SOC content and indicators derived from the fast wetting test of Le Bissonnais (MWD1, MA1). The role of SOM in promoting soil structural stability is well-know (Chan and Mullins, 1994; Fukumasu et al., 2022; Kong et al., 2005; Regelink et al., 2015; Six et al., 2004;

Tisdall and Oades, 1982), as SOM has long been recognised as one of the main binding agent in micro-aggregates (Dexter, 1988; Edwards and Bremner, 1967). The increase in SOC along a field gradient has been shown to decrease the wettability of individual aggregates from 3 to $5\,mm$ in diameter and of SOM-associated clay in the $<\,2\,\mu m$ fraction of soil (Chenu et al., 2000). This decrease in clay wettability might explain, in part at least, the higher structural stability under water of soils rich in SOC, with the slower wettability of macro-aggregates explaining for their improved resistance to slaking (Chenu et al., 2000)

and the slower wettability of clay decreasing its dispersive character (Chenu et al., 2000; Dexter et al., 2008).

In contrast, while the absolute clay content increases soil resistance to raindrop impact (supported by the positive correlation with MWD3 and MA3), it also tends to decrease soil structural stability under water. This supports the view that, for cropland





soils of this study low in SOC content on average, clay dispersivity and differential swelling are strong drivers of soil disaggregation in wet conditions. This is in agreement with the findings of Dexter et al. (2008) who found that, for soils from France and Poland, clay has a dispersive power in water that is reduced once complexed with SOM, with an average complexation potential of $1\,\mathrm{g}$ of SOM for $10\,\mathrm{g}$ of clay. This threshold value of $0.1$ for mass SOC:clay ratio was reported as pivotal between good and medium structural quality as estimated by field visual soil assessment by the CoreVESS method for 161 agricultural soils of Switzerland (Johannes et al., 2017) and for a large number of forest, grassland and cropland soils from England and Wales (Prout et al., 2020). Additionally, both Johannes et al. (2017) and Prout et al. (2020) found a linear increase in soil structural stability scores with increasing SOC:clay ratios in the range $1:13$ to $1:8$, suggesting that SOM has beneficial effects on soil structure beyond the threshold value of $1:10$ determined empirically by Dexter et al. (2008). We assume that that these results can be extrapolated to other temperate European soils under similar pedoclimatic conditions and clay mineralogy, as supported by the linear increase of QST indicator $W_{max}$-$W_{t0}$ with the SOC:clay ratio in the range $0.04 - 0.12$ (Fig. 3). This supports the idea that $W_{max}$-$W_{t0}$, as a predictor of the SOC:clay ratio, has the potential to evaluate the overall soil structural stability as measured in field conditions with visual soil assessment methods.

It is important to underline that the close linear relationship found between $W_{max}$-$W_{t0}$ and the SOC:clay ratio has probably no general character and was obtained here because cropland soils of the current study were sampled under standardised conditions of seeding and cover (winter wheat). It is very unlikely to find an identical relationship for the same soils under contrasting conditions of soil preparation, sampling dates or crop type, since soil structural stability doesn't only relate to the SOC:clay ratio but also to more or less dynamic external factors such as tillage, root and hyphae development, biological activity, etc... To sum up, we suggest that the SOC:clay ratio must be seen as a proxy for soil intrinsic 'potential' structural stability, with the threshold value of 0.1 being a reasonable target for SOM management at field and farm scales (Dexter et al., 2008; Johannes et al., 2017; Prout et al., 2020). On the other hand, $W_{max}$-$W_{t0}$ from QST curves provides a quantitative, direct measurement of the overall structural stability of a soil under a given set of conditions. Both parameters are therefore relevant in terms of appreciation of soil resistance to water erosion and structural damage by farm machinery.

## 4.2 The response of QST indicators to agricultural practices

One challenge for the interpretation of QST curves is the choice of the most suitable indicator(s) to assess overall soil structural stability in a given context. For the tillage and P-K mineral fertiliser trials, the choice of one indicator rather than another is not critical because indicators from QST curves are consistent with each other and with indicators from Le Bissonnais. In contrast, for the treatments of the organic matter trial, results are discordant between indicators from the start and the end of the QST curves. This originates from curves having different shapes according to the treatments. The FYM treatment resists better to disaggregation at the start of the QST, whereas RR is the best treatment against disaggregation under water at the end of QST curves (Fig. 4). This last result is counter-intuitive, since the FYM treatment has a higher total SOC content than the RR treatment. Nevertheless, similar results had already been reported on the same trial (Droeven et al., 1980), with the RR treatment resisting better than FYM to disaggregation by wet sieving. This result in conflict with total SOC content must be regarded in light of (i) the quality of SOM inputs and (ii) the frequency of SOM restitution. Buysse et al. (2013a) calculated that the FYM





and RR treatments receive on average similar amounts of C inputs, $472 \pm 82$ and $487 \pm 93\,\mathrm{g\,C\,m^{-2}\,y^{-1}}$, respectively. Over time, this amount of C input by farmyard manure application has led to an increase of total SOC content (about $12\,\mathrm{g\,kg^{-1}}$ for the FYM treatment) whereas for the RR treatment, an equivalent C input by green manure and residue restitution only allowed

to maintain SOC content to the initial level (about $10\,\mathrm{g\,kg^{-1}}$; Buysse2013a). A smaller SOC storage for a similar C input means a higher rate of mineralisation. The formation of water-stable aggregates under the effect of microbial decomposition of root biomass is a known process (e.g., Dufey et al., 1986). In the present study, the more microbially active, labile biomass from green manure and crop residues seems to have had a stronger impact on the later part of the QST curve, controlled by clay dispersion, whereas the more processed, stable biomass of the farmyard manure had more impact on soil resistance to

slaking. This is in agreement with an important contribution of root and fungal exsudates as well as microbial mucilages, known as critical binding agents in micro-aggregates (Dexter, 1988), on the reduction of clay dispersivity. Dispersive clay has been proved an important driver of soil erodibility (Brubaker et al., 1992; Czyż and Dexter, 2015). In contrast, soil resistance to slaking appears to be more related to the total content of SOC, with slaking having little effect on micro-aggregates $< 250\,\mathrm{\mu m}$ (Dexter, 1988). Another important point is the frequency of SOM inputs, with the FYM treatment receiving cattle manure once

every three years whereas the RR treatment receives an extra SOM input annually, in the form of green manure or chopped straw. The FYM treatment might have obtained better soil structural stability scores if sampling had occurred shortly after FYM application (the last FYM application almost two years before the sampling campaign).

For the tillage trial, reduced tillage (RT) improves soil structural stability regardless of the QST indicator (Fig. 6), even though some late-stage and global indicators (Slope$_{\text{max-300}}$, Slope$_{\text{max-600}}$, W$_{\text{end}}$ and AUC) tend to discriminate better between

tillage treatments. This result is consistent with an increase in both total SOC content and root biomass in the $2-7\,\mathrm{cm}$ topsoil (RT: $42 \pm 19\,\mathrm{mg}$ of root biomass for $100\,\mathrm{cm^3}$; P: $31 \pm 16\,\mathrm{mg}$ of root biomass for $100\,\mathrm{cm^3}$, p=0.168). The gradient of concentration of SOC and nutrients from the surface soil under RT is a known effect once vertical dilution by ploughing is stopped (Chervet et al., 2016; D'Haene et al., 2009; Luo et al., 2010). This higher nutrient content in the topsoil may explain for the higher root density. Whereas a higher root density is known to play a key role in soil macroaggregation, a higher root density

and SOC content in the topsoil also advocate for a higher biological activity. This is in line with a better microaggregation and a better performance of indicators from the end of QST curves under RT, related to a better resistance to clay dispersion. This is supported by the fact that the relationship between QST indicators and the amount of root biomass was relatively poor for QST indicators from the start of the curves (Fig. 7a-c) and increased for the later ones (Fig. 7d-f). Overall, results from the OM and the tillage trial support the view that living and labile biomass plays an important role in decreasing clay dispersion. This

result is in agreement with the fact that labile biomass from green manure and crop residues has more effect than composted farmyard manure on the reduction of clay dispersion in the OM trial.

For the P-K mineral fertiliser trial, the working assumption that KCl application might decrease soil structural stability due to the presence of destructuring chloride anions (Paradelo et al., 2016) was not verified. This is probably due to the fact that the disaggregating effect of $\mathrm{Cl^-}$ is relatively short-lived after application of KCl, with $\mathrm{Cl^-}$ being lixiviated downwards over

time with water fluxes. Since the last KCl application occurred in the summer of 2016, the absence of residual disaggregating



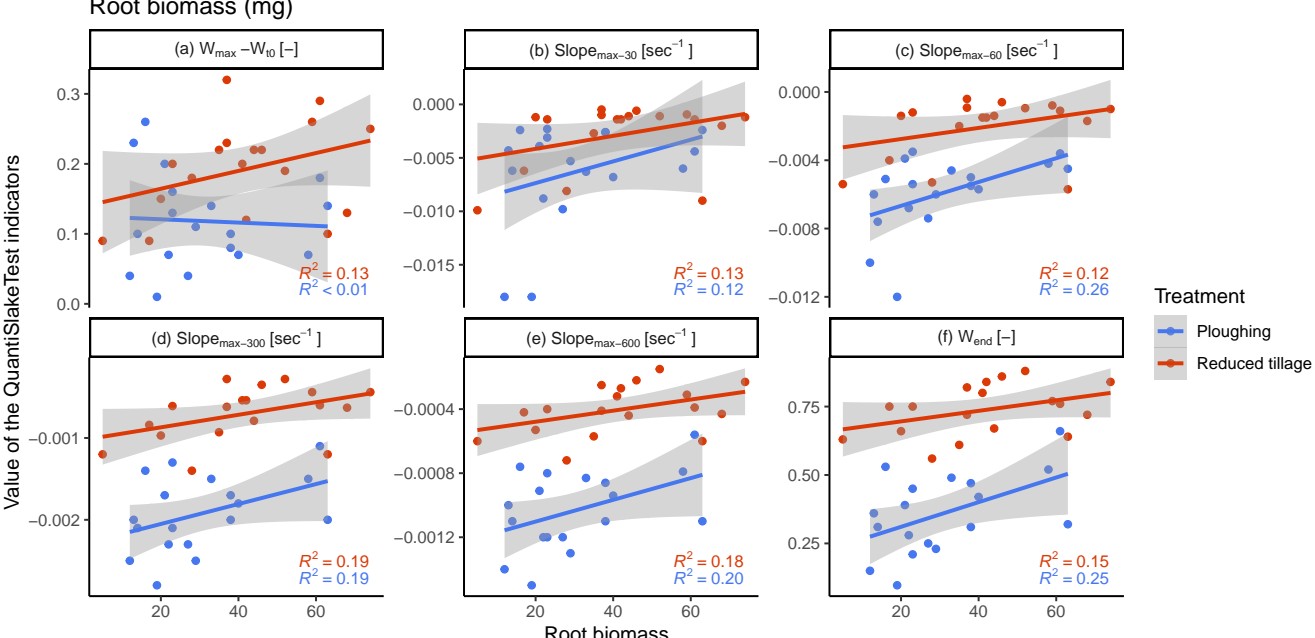

**Figure 7.** Scatterplots for six indicators ($W_{max}$-$W_{t0}$, $Slope_{max-30}$, $Slope_{max-60}$, $Slope_{max-300}$, $Slope_{max-600}$, $W_{end}$) from various individual QST curves from the tillage trial against root biomass (mg) collected in the basket after running the QST.

effect from $Cl^-$ anions is not surprising. The beneficial effect of K fertilisation on crop production and restitution of organic matter to soil might also have further counteracted a potentially negative short-term effect of $Cl^-$.

### 4.3 Advantages, limitations and perspectives of development of the QST test

The main strength of the QST relies on its simplicity, as the test is rapid to run and doesn't require expensive equipment or
laboratory consumables (distilled water is actually the only consumable required). QST measurements can therefore easily
be repeated several times for one single plot, to improve the robustness of the result by decreasing both the impact of field
microsite heterogeneity and of analytical error. Another point of interest is that the QST works on a large structured soil
volume (Kopecky cylinders of $100 \, cm^3$ in the present study) whereas most traditional methods apply to a certain amount of
small aggregates from a soil previously gently crumbled by hands (Le Bissonnais, 1996; Edwards and Bremner, 1967; Hénin
et al., 1958; Kemper and Rosenau, 1986; Yoder, 1936). The use of a large soil volume may increase the representativeness of
the soil sample while decreasing the risk of bias introduced by the selection of soil aggregates from a given size fraction (the
test then neglects the properties of the soil fraction of inferior or superior equivalent diameter). To promote the adoption of the
method by a wide public, an opensource R package 'slaker' (Vanwindekens and Roisin, 2022) including a web application is
currently under development for QST data acquiring, management and analysis, including the calculation of relevant indicators
and statistics from the curves and the provision of some keys of data interpretation. Therefore, the QST has a strong potential



for adoption by a widespread community of end-users from soil science laboratories to farmer organisations with no or little expertise in the measurement of soil properties.

Beyond its simplicity and its large adoption potential, the dynamic character of the test is another strong point, since a high density of information stands in one single curve. On the one hand, it has the advantage to provide at once information on the
two main mechanisms of soil disaggregation under water (slaking and clay dispersion), and offers the possibility to calculate a diversity of indicators for curve interpretation, focusing either on one specific mechanism of disaggregation (e.g. $Slope_{max-60}$ for slaking and t95 for clay dispersion) or on the overall structural stability of soil ($W_{max}-W_0$, $W_{end}$, AUC). In this regard, the strong linear relationship between $W_{max}-W_0$ and SOC:clay ratio (which can be considered as a proxy for the estimation of the 'potential' structural stability of a soil, Johannes et al., 2017, Prout et al., 2020), supports the view that $W_{max}-W_{t0}$ is relevant to
evaluate the overall soil resistance to disaggregation in field conditions. This kind of indicator may therefore be more relevant for the overall appreciation of soil structural stability than indicators related to one specific mechanism of soil disaggregation.

On the other hand, one can stand that the focus of the test is not clearly defined in terms of mechanism of disaggregation as the responses of soil to slaking and clay dispersion overlap. In that respect, a perspective of improvement of the test is to get rid of the interference of air bubbles leaving the sample in the early stages of the QST. This could be reached by measuring
soil mass leaving the basket in addition to that remaining in the basket. By removing this 'air release' variable, we hypothesise that the curves would result only from the overlapping effects of slaking and clay dispersion, which would reduce the number of explanatory variables to two and allow for a successful curve modelling, in order to decompose the curves regarding the respective contribution of slaking and clay dispersion. Another challenge is that information from one indicator can somehow be contradictory with the information from another indicator from the same curve, as shown in the present study for the OM
trial. To tackle this issue, the construction of a QST library to better assess the response of QST indicators to soil management practices and soil properties is necessary to objectify the choice of suitable indicators from the curves and provide keys for their interpretation.

In its current state, the test doesn't provide information on the size of aggregates surviving disaggregation under water, which is of interest to predict soil susceptibility to water erosion. Nevertheless, measurement of residual aggregate size distribution
with classic sieving method would decrease the convenience of the test. Coupling the test with a particle size analyser by dynamic image analysis is another perspective of development for rapid determination of size distribution of particles leaving the basket. As it stands, the test doesn't provide any information on soil resistance to raindrop impact. However this point is not critical as soil resistance to raindrop impact is routinely estimated by pedotransfert functions using pH, SOC and clay contents as input variables (Remy and Marin-Laflèche, 1974).
To sum up, the QuantiSlakeTest has many strengths, and many perspectives of development exist to tackle the existing issues and better exploit the QST curves.



## 5   Conclusions

In this work, we propose a new method to evaluate soil structural stability, the QuantiSlake Test (QST). It consists in the dynamic weighting of a structured soil sample under water and the calculation of several indicators from the curves to evaluate soil structural stability. The QST presents several advantages. First, it is rapid to run and works with structured soil samples of large size, which improves the representativeness of the sample and allows for multiple field repetitions. Second, the QST doesn't require expensive equipment or laboratory consumables. Third, a high density of information stands in one single curve, with the possibility to extract information either on specific mechanisms of soil disaggregation (slaking and clay dispersion), or on the overall structural stability of soil. Several perspectives of improvement of the QST are under study, such as the decomposition of the overlapping mechanisms of soil disaggregation by curve modelling and the development of an online program for data management, automated calculation of indicators and statistics from the curves and providing keys of interpretation. Therefore, the test has a strong potential for adoption by a widespread community of end-users from soil science laboratories to farmer organisations with no or little expertise in the measurement of soil properties.

In the present article, we show that the early mass loss under water is mainly related to slaking, whereas after soil saturation with water, clay dispersion becomes the dominant process of soil disaggregation. We found that soil resistance to both slaking and clay dispersion is closely related to the SOM status of soil, well-captured by the SOC:clay ratio. From our results, we confirm the validity of the SOC:clay as a proxy for the estimation of soil intrinsic 'potential' structural stability, with the threshold value of 0.1 being a reasonable target for SOM management at field and farm scales for the soils of central Belgium (Dexter et al., 2008; Johannes et al., 2017; Prout et al., 2020). On the other hand, we propose that the early increase in soil mass systematically recorded shortly after introduction of soil in water ($W_{max}$-$W_0$) when running QST on cropland soil provides a quantitative measurement representative of soil structural stability as it stands in field conditions. Both parameters are therefore relevant in terms of appreciation of soil resistance to water erosion and structural damage by farm machinery.

Beyond the absolute amount of SOC for a given level of clay, the response of QST indicators to agricultural management practices highlighted that the quality of SOM inputs affects both SOC storage and soil resistance to disaggregation. In the organic matter trial, for similar total SOC inputs, farmyard manure favoured the total SOC content and had the best soil resistance to slaking whereas green manure and restitution of crop residues improved soil resistance to clay dispersion the most. We conclude that living and labile biomass is more efficient in decreasing clay dispersivity whereas soil resistance to slaking relates to total SOC content. This underlines that the choice of indicators for the interpretation of QST curves must be done with great caution, as indicators from the start and the end of the curve may lead to conflicting conclusions.

*Code availability.*  R-package - slaker - Analysing the data of QuantiSlakeTest approach. R-package and Web Application, https://gitlab.com/FrdVnW/slaker ; Notebook with codes, figures and tables - qst-openscience, https://frdvnw.gitlab.io/qst-openscience/



*Data availability.*   Data repository in the SlakingLab community on Zenodo

https://doi.org/10.5281/zenodo.7142458

520   *Code and data availability.*   Full git repository - qst-openscience

https://gitlab.com/FrdVnW/qst-openscience

*Video supplement.*   A visualisation of the QuantiSlakeTest, comparing two contrasted samples (tillage / conservation tillage) and curve generation ; Tuto slake 1, in french ; Tuto slake 2, in french

*Author contributions.*   FV and BH planned the sampling campaign and the measurements, analysed the data, wrote the manuscript draft and
525   reviewed and edited the manuscript.

*Competing interests.*   The authors declare no competing interests.

*Acknowledgements.*   Many thanks to Dr Ir Christian Roisin, whose questions and ideas initiated the development of our approach. Thanks to the first students who used the QuantiSlakeTest during their training or in their thesis: Vincent Gaucet, Cyril André, Mathieu Dufey, Clément Masson, Fanny Lizin. We want to thank peers for fruitful exchanges around our approach and the early results of its application during
530   conferences. This study is part of the PIRAT project (CRA-W) and supported by the action plan BIO2030 (CRA-W / Wallonie).





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
