# Peer review of "The QuantiSlakeTest, measuring soil structural stability by dynamic underwater weighing"

_EGUsphere, 2022_

## Referee Comment (RC2)

[referee-annotated manuscript omitted]

---

## Author Response (AR1)

**Comments reviewer 1 (submission 1) and answers from authors**

*N.B. : We update our answers here based on the last version of the revised manuscript, amended after receiving the comments from Reviewer 2. This is shown in yellow in the following text.*

The manuscript presents a very interesting study on a new laboratory method to measure soil structural stability. The subject is well introduced, the statistical analysis is thoroughly executed using a nice and robust dataset.

The manuscript is very well written, well organised, interesting and very clear. There is also a very interesting discussion on the mechanisms controlling soil disaggregation.

the paper addresses a relevant scientific question within the scope of SOIL by providing new soil health indicators determined by a new and simple method. The results are compared with results obtained using a reference method (MWD).

Moreover, the authors paid particular attention to the publication of FAIR data and codes.

- *We appreciate the kind feedback of Mr Saby and we are very grateful for his time and helpful comments.*

I have very little comments

- **[X] In the summary, l8, it is not clear that you use some indicators based on the curves to the derive information on soil structural stability. Maybe you could add one sentence to specify this point.**

    o *The term "indicators" was mentioned earlier in the paragraph, in the objectives of the paper, but we added details also in l8, for more clearly point that these are calculated from the curve at some typical dynamic step:*

    o *"For each plot, QST indicators derived from curves (e.g. relative increase or decrease in soil mass, disaggregation speed, time to reach a threshold value of mass loss …) were compared to the results of the three tests of Le Bissonnais, targeting specific mechanisms of soil disaggregation."*

    o *UPDATE  in the revised version : "For each plot, indicators calculated from QST curves (e. g. total relative mass loss, disaggregation speed, time to meet a threshold values of mass loss, ...) were compared to the results of the three tests of Le Bissonnais (1996), used as a reference method for the measurement of soil aggregate stability."*

- **[X] L89, 'in field conditions', it is not clear to me. SLAKES mobile application should be use in a lab or a room.**

    o *The strength of SLAKES application relies on the rapid acquisition of data on soil structure with little equipment needed (the measurement can be performed inside but also outside in absence of rain or wind) → we propose to remove "in field conditions":*

- o *"Recently, the SLAKES mobile application provided encouraging results as a tool for rapid data acquisition on soil structure . The test relies on image recognition to measure…"*

- **[X] L100 there is a repetition of the definition of the QST acronym?**

  - o *Yes, a mistake, we adapted:*

  - o *"In this work, we evaluated the performance of a new, simple test to evaluate soil structural stability, named QuantiSlakeTest (QST), a quantitative approach of the slake test."*

  - o ==*UPDATE in the revised version : "In this work, we evaluated the performance of a new, simple test to measure soil structural stability, named QuantiSlakeTest\ (QST). It is a quantitative approach of the slake test, a visual qualitative test to illustrate the impact of soil management practices on soil structure. It consists in the dynamic weighing of a structured soil sample suspended in demineralised water. This approach has the advantage of being simple, rapid and dynamic, therefore providing a high density of information throughout the process of soil wetting and disaggregation under water.*==

- **[X] Is Table 1 published on a data repository? If yes, add the DOI in the caption and the text.**

  - o *The data were available via the "gitlab" repository (https://frdvnw.gitlab.io/qst-openscience/). But not in a real "data repository". We added this table in the data repository linked to the "SlakingLab - users, data and codes for running QuantiSlakeTests" community on Zenodo. Reference and doi : Vanwindekens, Frédéric M., & Hardy, Brieuc F. (2022). QST - open data of the article Vanwindekens & Hardy (2022) - table 1 soil properties (1.0) [Data set]. Zenodo. https://doi.org/10.5281/zenodo.7405113*

  - o *As suggested, we will add the doi link in the text, just after the ref to the table and in the caption of the table 1.*

- **[X] L230-240, the choice of the different indicators could appear arbitrary and in particular for the time (ii) or the threshold value (iii). I don't think it is discussed. More generally, the use of the QST curves could be discussed in terms of statistics. You may use functional data analysis or curve modelling instead. Fajardo et al. use a Gompertz model and interpret the parameters of the curves. Maybe this point could be mentioned for future research. I think there is only one short sentence in the conclusion (l495).**

  - o *This is a very good point, Thank you for this comment. Actually, curve modelling has been tested (we used double exponential models) but the increase in soil mass occurring at the start of the test makes modelling complicated and model parameters correlated poorly with MWDs and soil properties. A perspective that we see is to measure the mass of soil leaving the basket additionally to the mass of soil in the basket. This second measurement would have the advantage to get rid of the interference of air bubbles leaving the basket and therefore limiting the number of factors controlling soil mass evolution under water. At the moment our device doesn't allow such a measurement. We believe that such a development has the potential to improve greatly curve interpretation regarding the underlying mechanisms of soil disaggregation*

- **[X] Figure 4 and 5 I think we are missing some results on the MWD here to be able to compare.**

  o *We think reviewer point figures: 4 and 6 (same kind of figures, with boxplot). We structured the results of our paper in three main parts: i) comparison QST <-> Le Bissonnais ii) indicators against soil properties (using QST & LeBissonnais) iii) then comparing the treatments of the LTE against soil structural stability, but with a focus on the QST indicators. The two first parts were there to evaluate the QST curves in comparison with recognized ones (Le Bissonnais), while for the comparison of contrasting farming practices, we used only the QST as the central approach of the paper. That's why it did not come into our mind to add boxplot with MWD indicators.*

  o *We think also that the 2 graphs are already complete. We already made a "sub-selection" of most relevant QST indicators.*

  o *Nevertheless, we found the proposition of the referee and we propose to add 3 graphs in supplementary materials. We will add a reference to these graphs in the text of this part.*

  o ==*UPDATE  in the revised version : 2 graphs are added in the main manuscript, linked to 2 main trials Organic Matter and Tillage. The graph with PK -fertilisation trial is available in the online supplementary report.*==

- **[X] L286 and Fig 3, yes there is a clear relationship but also there is a residual variability. I think this should be acknowledged in the text.**

  o *In this part of the results, we focus the presentation of our data on correlation coefficients between variables from soil properties and from indicators. Our aim is not to produce a predictive approach with a model. We observe a standard deviation in the repeated test of a same plot, that could be explained by local soil conditions of the sampling sites (eg. slopes, presence of roots, or earthworms' galleries). We will acknowledge that in the new version of the text.*

  o

- **[X] L344 some QST indicators. It is possible to be more specific?**

  o *We will specify in the text:*

  o *"Therefore, we think that that QST indicators $Slope_{max-60}$ might be more specific to slaking than the fast-wetting test of Le Bissonnais."*

  o ==*UPDATE in the revised version : We therefore advocate that indicators from the initial stage of the curve, like $Slope_{30-60}$, may provide information much more specific to slaking than indicators from the fast wetting test of Le Bissonnais, lasting ten minutes, which largely exceeds the time during which slaking is the dominant driver of disaggregation.*==

- **[X] L386 Add reference to figure 3 for clarity?**

  o *Added:*

  o *"It is important to underline that the close linear relationship found between $W_{max}$-$W_{t0}$ and the SOC:clay ratio (see Fig. 3)) has probably no general character and was obtained here because cropland soils of the current study were sampled under standardised conditions of seeding and cover (winter wheat)."*

- **[X] L410: something wrong with the reference.**

  o *Yes, a mistake in the manuscript (omission of a LaTeX command). We adapted:*

  o *"… allowed to maintain SOC content to the initial level (about 10 g kg−1; Buysse et al., 2013b). A smaller SOC storage for a similar C input means a higher rate of mineralisation. The formation of water-stable aggregates under the effect of microbial decomposition of root biomass is a known process (Dufey et al., 1986). In the present study, the more microbially active, labile"*

- [X] **L450 the use of a large soil volume facilitates the soil sample preparation compare to MWD.**

  o *We agree that the soil sample preparation of the QST is easier compare to MWD, but we don't think that the sample size is the cause. We will add in the paper, in the beginning of this paragraph:*

  o *"The soil sample preparation is far easier than approaches based on MWD as, after removed from the steel cylinder, the sample is dried "as is" and used "as is" in the QST."*

  o *If soil sample preparation is shorter, we don't see it as critical because clods crumbling by hand and soil sieving to 3-5 mm is neither complicated nor time consuming. To our point, the main advantage is the absence of bias compared to sieving that selects aggregates of a certain size, neglecting particles < 3mm that can represent a large fraction of a soil that is frequently tilled*

- **[X] L500 this sentence is right only for soils of central Belgium for the moment.**

  o *We agree (and we are very interested to test the approach in other pedological context). We will adapt this paragraph of the conclusion as following:*

  o *"For cropland soils of the silt loam region of central Belgium, we show that the early mass loss under water is mainly related to slaking, whereas after soil saturation with water, clay dispersion becomes the dominant process of soil disaggregation. "*

  o *UPDATE in the revised version : In the present article, the QST was applied to 35 agricultural soil samples from three long-term experiments in the silt loam region of central Belgium.  For these soils, the early mass loss under water was mainly related to slaking, whereas after soil saturation with water, clay dispersion and differential swelling became the dominant processes of soil disaggregation. We found that soil resistance to disaggregation is closely related to the SOM status of soil, well-captured by the SOC:clay ratio. From our results, we confirm the validity of the SOC:clay as a proxy for the estimation of soil intrinsic 'potential' structural stability for the soils of central Belgium, as it correlated strongly with QST indicators.*

**Comments reviewer 2 (submission 1) and answers from authors**

First of all, apologies for the delayed response.

This article presents a relevant, novel and worthwhile adaptation of an existing soil structural stability test, and applies it to different case studies, demonstrating its potential for routine evaluations within the context of soil (structural) quality assessments. It certainly deserves to be published. Nevertheless, in my opinion, the introduction needs to be revised, a number of issues have to be clarified (see details below) and additional indicators derived from the QST curves could be tested (without requiring additional measurements).

*Authors : Dear Reviewer,*

*First we would like to thank you about the time spent for a serious, in depth revision of our work. Most comments appear to be relevant and will help to strengthen the manuscript.*

**Introduction**

- The introduction is quite lengthy. There is a lot of interesting information, but it does not fully serve the purpose of the paper. Part of the difficulty stems from the fact that the authors were not able to decide whether the main focus of the paper is about a new tool or about understanding the effects of various practices on their novel indicators. To me, the paper is first and foremost about a new technique to quantify soil disintegration during immersion, which is then applied to a variety of soil samples from different cropping systems from loess soils in central Belgium. The focus of the introduction should therefore be on existing techniques and why another method like the QST is relevant (L78 and subsequent). L23-49 do not really seem relevant for introducing the QST.

*Authors : Thank you for this comment. We agree that the main originality of the work stands in the new method of aggregate/structural stability measurement by soil immersion under water. Nevetherless, it is also clear that a new method can hardly be evaluated properly without an appropriate case study that offers a range of conditions and outputs for the test.*

*The whole manuscript, from introduction to discussion, is written accordingly, with a balance between 1) QST method description and evaluation and 2) application of the QST to silt loam Luvisols (having a fragile structure) subject to contrasting soil management practices. The introduction respects this duality, with one part presenting the soil and agricultural context of the loess belt of central Belgium and the other one focussing on soil aggregation and aggregate stability measurement.*

*We don't want to lose this duality because it represents the guideline and the genesis of our work, and that limiting the text to the QST method only would be a substantial loss of information, important to evaluate the method itself. We believe that the agronomic outputs of the manuscript can be of interest for many readers of SOIL and will not prevent other readers to put the focus on the methodology.*

- It is also not obvious to me how relevant it is to discuss the hierarchical levels of soil structure, except that the samples should be sufficiently large to encompass all levels so as to be representative of the soil.  The fact that the authors do not revert back to these concepts in their discussion clearly indicates that this information is of limited relevance for this paper.

*Authors : We understand the point raised by reviewer 2 but we believe that reminding some key elements of soil aggregation is not useless at all for such a paper. First, some critical advances of research on soil aggregation date from the eighties and even earlier. Therefore, several milestone papers are not easily accessible online. Second, the ultimate goal of the test is to provide a practical, cheap and effective solution for soil stability measurement for a wide community of end-users, including those with little expertise in soil analysis. We are convinced that many of the readers of this text and final users of the test will greatly benefit from such information, in order to provide them a clearer idea of what stands behing the concept of soil aggregation (actually the feedbacks that we have received on the paper by now support this point).*

- In English, the usual term is 'aggregate stability', which in the case of Le Bissonnais is indeed what is being measured (3-5 mm aggregates). It may be worthwhile discussing the distinction the authors make between  aggregate stability and structural stability (if any).  I would argue that QuantiSlake = structural stability, whereas Le Bissonnais = aggregate stability …

*Authors : We thank you for this important remark. Indeed we think that it is useful to stick to "soil structural stability" rather than to "aggregate stability" because we work on soil volumes exceeding the size of soil aggregates. This information will be added in the M & M in the revised version of the text. We propose the following complement of information l. 243 :*

*" Whereas Le Bissonnais (1996) and many reference methods measure the stability of soil aggregates, the QST rather measures soil structural stability, as it works on 100 cm³ soil volumes rather than on soil aggregates. Therefore we will stick to "soil structural stability" when referring to QST measurements."*

*We will also check carefully to avoid the use of "soil structural stability" for other reference methods measuring "soil aggregate stability"*

**Specific comments**

- L9 : this is not fully correct. see comments further down on the Le Bissonnais test.

*Authors : Agreed, thank you for this remark. Will be revised all over the text cf. Comment l.211-212.*

- L32 I suppose liming is also an important practice in these systems, that has contributed to changes in pH and base saturation. Liming does not fall under 'organic and mineral fertilizer'. It is a soil amendment.

*Authors : will be replaced by "organic and mineral fertilizers and amendments"*

- L35-37 please add reference

*Authors : reference : Goidts & Van Wesemael (2007)*

- L42 add reference for the 'critical value of SOC content'

*Authors : reference : Meersmans et al (2011)*

- L47 '…. According to farmers ….' and scientists !

*Authors : we propose to remove "According to local farmers" in the revised version so the statement is more general*

- Starting L46 (till L65), the text becomes somewhat confusion. The paragraph starts about conservation tillage, then switches to erosion, then switches to aggregate stability and the hierarchical structure of aggregates : these are a lot of different ideas for a single paragraph. In general, 1 paragraph = 1 idea.

*Authors : This is where the transition between 1) the soil and agricultural context and 2) soil aggregation and soil aggregate stability measurement occurs. The order of information "soil erosion" > "soil aggregate stability" > "soil aggregation theory" doesn't sound inappropriate to us, as i) the issue of soil erosion is the driver of our research; ii) Soil aggregate stability measurement is the most common proxy to assess soil erodibility and iii) the theory of soil aggregation is key to understand the properties of soil aggregates and mechanisms of soil aggregation and disaggregation.*

*In the revised version, we propose to make the transition smoother and to reorganize the paragraphs (l. 53):*

*"..., soil aggregate stability is often used as an indicator of soil erodibility (Barthès & Roose, 2002).*

*The process of soil aggregation is critical to understand the factors controlling soil aggregate stability. The theory of aggregate hierarchy of Hadas (1987)..."*

- L57 Ca++ … and Mg++ (as a rule, divalent cations are much more effective than monovalent cations)

*Authors : the sentence will be completed accordingly*

- L68 The mechanisms listed here are relevant for soil sealing / soil erosion studies.  I would argue that in a broader context, mechanical breakdown also occurs as a result of mechanical stresses exerted during tillage operations or traffic.

*Authors : Thank you for this. "The resistance of soil to mechanical breakdown also improves resistance to soil compaction due to traffic on the field" will be added l. 71.*

- L86-87 : the method of Koestel is not really about measuring aggregate stability; Likewise, the use of VIS-NIR does not allow to measure stability : it is based on correlations, and therefore will always require reference methods.

*Authors : We agree with this remark, we propose to split the information, with spectroscopic techniques providing information on soil structure mentioned in a separated paragraph:*

*"Recently, the potential of some non-destructive methods on the evaluation of soil structure and aggregation has been revealed, such as..."*

*Accordingly, the information about the SLAKES application (l. 88-90) will be moved up and attached to the previous paragraph*

- L90-93 : there is some confusion here in the text between 'aggregate size distribution' (or level of aggregation) and 'aggregate stability'.  Furthermore, the "profil cultural" (and other methods listed here) do not seek to measure aggregate stability.

*Authors : We propose to remove from l.91 to 93 about the field methods for the evaluation of soil structure, for the sake of clarity.*

- L95 : not sure I understand why topography and climate would influence the choice of method; please clarify

*Authors : The idea is that topography and climate are key controls of erosion risk. The amount, frequency and intensity of precipitations and length and steepness of the slope will control erosion risks as much as soil intrinsic erodibility. Therefore, in a flat area in the temperate zone, soil aggregate stability in wet conditions may provide useful information on the risks of soil sealing and crusting or compaction whereas in other contexts the mechanical strength of soil, or a rainfall simulator approach may bring more relevant information.*

- L108-116 : can be deleted, as it will be repeated in materials and methods / discussion sections

*Authors : Actually we believe that l. 108-112 are absolutely necessary as they present the general approach of the work, in line with the objectives presented l. 104 – 108. We propose to remove the text from l. 113 to 116 (indeed somehow redundant with information within M&M)*

- L121 explain abbreviation : CRA-W

*Authors : Abbreviation is defined at first use (l. 109). For information, CRA-W refers to "Centre wallon de Recherches Agronomiques", with letters in a confusing order due for historical reasons …*

- L123 : round off to the nearest mm; the decimal really doesn't provide useful information

*Authors : We fully agree, this will be done (3 significant digits)*

- L148 latin names are provided here but were not given in the previous section when the crops were first named; please correct

*Authors : we will check carefully that the Latin name is given for each plant species at first use in the revised version.*

- L153 "Complete random block with split plot design" : doesn't sound quite right. 'split-plot' is not compatible with 'completely randomized' because by definition, at least one treatment (the 'split') is not fully randomized.  Please clarify.

*Authors :*

*Some sources indicate that the split-splot design is compatible with randomized complete blocks : Dagnelie P. (2012), Principes d'experimentation. Planification des expériences et analyse de leurs résultats. See. Chap 7.*

*After verification, the design of the tillage trial is more correctly described as a Latin Square with one studied factor (the four tillage treatments) and one controlled factor (the 4 blocks) without repetition (sensu stricto). We therefore propose the following formulation:*

*"The trial includes four tillage treatments repeated four times, following a Latin square design with the blocks aligned in a row."*

- L165 sampling occurred in April 2019 but no fertilizer was added since 2016 ? Please check

*Authors : P-K fertilisers are applied once per rotation, before the sugar beet, which is the starter of the three-year rotation. The last application before sampling occurred after the 2016 harvest of barley in summer 2016. The next application occurred after the 2019 harvest, so after sampling.*

- L171 I'm not aware that chlorides can affect soil structure in soils dominated by permanent charge, but potassium (monovalent cation) definitely does!

*Authors : we propose the following adaptation: "on the potential effect of contrasting levels of KCl application on soil structural stability (cf Paradelo et al 2016)"*

- L198 : did you replicate measurements for each plot ? I believe Le Bissonnais recommends 3-5 replicates per plot (and per test).

*Authors : Indeed 3 to 5 replicates are recommended by Le Bissonnais et al. (1996) but here we didn't make any replication within one single plot because the soil sampling area was very limited (1 m²) and because we relied on field repetitions (true repetitions decrease the relevance of pseudo-replication). Moreover, since the Le Bissonnais method works on 5-10 g of aggregates 3-5 mm in diameter, the result is already an average value for many small aggregates.*

- L200 : the mechanisms involved in the test are not presented correctly; see comment L211 (below)

*Authors : We agree that across the document we sometimes make the following shortcut : fast-wetting = slaking. This statement must indeed be qualified and we will revisit the text accordingly. Nevertheless, in the methodology, the initial formulation seems correct (fast wetting of dry soil aggregates in water aims to test first and foremost their resistance to slaking, even if clay dispersion and differential swelling may also play a disaggregating role after 10 min under water).*

*We propose the following formulation: l. 200 "The first test consists in fast-wetting soil aggregates in water, exacerbating the effect of slaking"*

*About description of the third test, see next comment*

- L203 : the whole point of rewetting the soil using alcohol prior to shaking in water is also to minimize slaking (and swelling and dispersion)

*Authors : Indeed. We propose to complete the sentence l. 203 "...to test their mechanical strength while minimizing slaking, differential swelling and dispersion"*

- L203 : No sieving at 50 µm during the immersion phase in ethanol ?

*Authors : we followed rigorously the norm ISO FDIS 10930:2011. After each treatment, remaining aggregates were transferred to the 50 µm sieve. The < 50 µm fraction is not recuperated for any of the three test (it is calculated by difference between initial weight and fractions remaining on the sieves)*

- L205 : Start a new paragraph  at 'Two main indicators …'

*Authors : We agree and will adapt the manuscript in this way.*

- L209 : how strongly correlated are the MWD and MA indicators ? Is it worth considering both ?

*Authors : Both indicators are recommended by the norm. They are generally positively correlated for test 1 and 3 but not much for test 2. MA 3 is generally more (negatively) correlated to QST indicators than MWD3 (see Table 2). We propose to add the correlation matrix in supporting information. We propose to remove MAs from Table 2 since we make little use of it. Please fin the correlation matrix here below*

|  | MWD 1 | MWD 2 | MWD 3 | MWD 1 - MWD 2 | MA 1 | MA 2 | MA 3 |
|---|---|---|---|---|---|---|---|
| MWD 1 | 1.000 | 0.612 | 0.053 | 0.319 | 0.872 | 0.317 | -0.120 |
| MWD 2 | 0.612 | 1.000 | 0.184 | 0.945 | 0.505 | 0.637 | 0.003 |
| MWD 3 | 0.053 | 0.184 | 1.000 | 0.199 | 0.095 | 0.466 | 0.828 |
| MWD 1 - MWD 2 | 0.319 | 0.945 | 0.199 | 1.000 | 0.243 | 0.632 | 0.053 |
| MA 1 | 0.872 | 0.505 | 0.095 | 0.243 | 1.000 | 0.372 | -0.114 |
| MA 2 | 0.317 | 0.637 | 0.466 | 0.632 | 0.372 | 1.000 | 0.398 |
| MA 3 | -0.120 | 0.003 | 0.828 | 0.053 | -0.114 | 0.398 | 1.000 |

- L211-212 : this way of presenting the tests is not strictly correct. rapid wetting involves slaking, dispersion, but also differential swelling! Slow wetting involves differential swelling and dispersion. The mechanical breakdown test seeks to minimize slaking, swelling and dispersion, but it is not obvious that this really mimics drop impact.  From the discussion, it appears that the authors are aware of all this, so why present the three test in such a caricatural way ?

*Authors : As stated earlier, we agree with this comment and we are aware of this. The text will be revised accordingly (see comment l. 200). The link between the three tests and the mechanisms of soil disaggregation will be qualified in the introduction and we will stick to*

*the names of the three tests, "fast wetting", "slow wetting" and "mechanical breakdown" in the other sections.*

- Table 1 : please specify the upper and lower limits for the different particle size fractions, as this can be different from one country to another

*Authors : We used the following limits based on NF ISO 11464 :*

- *Sand (coarse, > 200µm)*
- *Sand (fine, 50 µm - 200 µm)*
- *Silt (coarse, 20 µm - 50 µm)*
- *Silt (fine, 2 µm - 20 µm)*
- *Clay (< 2 µm)*
  - o *As fine/coarse sand and silt data are not used in the manuscript we will simplify the table by removing them. In the revised version of the manuscript will only stand total sand, silt and clay:*
    - *Sand (50 – 2000 µm)*
    - *Silt (2 µm - 50 µm)*
    - *Clay (< 2 µm)*
- L225 not very clear. According to L220, the first measurement should be within less than 1 second after plunging the sample into the water. What does 'right after buoyancy' refer to ? How do you determine the time to buoyancy? Maybe this could be illustrated graphically.

*Authors : Indeed an illustration speaks from itself. Actually the graphs as shown on figure 2 and figure 5 only show soil mass under water after Archimedes upward buoyant force (once soil sample is completely immersed) --> if you keep soil mass data before and during immersion, you see the big drop from Archimedes buoyancy.*

*We will produce a didactic graph in the new proposal. Here a rapid "raw" graph. We use simple mathematical approaches based on derivatives ('primary', 'secondary') for finding the first minima. This point id considered as the "T0" and points before it are deleted (soil sample in the air, or partly in the air). The final graph that we will propose will complete the fig. 2 of the submitted version (graph with indicators).*

[Figure]

**Slake - 382**

[Figure]

- L229: start new paragraph at 'Several ….'

*Authors : We agree and will adapt the manuscript in this way.*

- L229 If I understand well, this maximum mass is going to depend on the relative rate of disintegration vs. relative rate of wetting, as well as mass of the sample (corrected for buoyancy based on the volume of the solid phase). Doesn't this also contain information?

*Authors : As explained in our previous comment, we deleted information before and during immersion (the buoyancy). Soil mass is then normalized according to the maximum value reached (WMax=1) after the considered T0 (first minimum), so that soil mass is expressed as a relative mass.*

*If you now look at figure 2, you see how soil mass generally behave for the soils of this study: soil mass increases due to wetting (due to water filling in porosity) and reaches a maximum (WMax = 1) before decreasing (once mass loss due to disaggregation becomes dominant compared to mass gain by wetting).*

*Going back to Archimedes buoyancy: indeed, mass drop due to immersion certainly contains extra information that we didn't exploit in the present manuscript.*

*To clarify, we propose the following reformulation :*

*l.225 - … : "soil mass drops due to Archimedes upward buoyant force (data not shown). The first value of soil mass under water is defined as the time 0 (t0) of the QST test (soil mass before and during immersion is removed from the graph and not shown here). The graph on Figure 2 illustrates how soil mass behaved under water for the soils of the present study. In the initial phase, soil mass generally increases due to water filling porosity. After a few seconds or minuts, the soil mass reaches a maximum (Wmax) before decreasing, once mass loss due to disaggregation becomes dominant compared to mass gain by wetting. Soil mass was normalised according to Wmax, so that mass value vary between 0 and 1. …"*

*l. 231 --> 241: "mass loss" will be changed to "relative mass loss"*

*Authors : We don't fully understand the question. I will try to clarify. All our graphs and indicators are based on a relative weight. As we used the Wmax as reference (denominator), the relative weight at "Max" is 1 (see also answer for l240).*

- L234-236 this has to be better explained.  The sentence seems to indicate that it is the slope of the curve at t= 30 sec (with t0 taken as tmax), but this is not what Fig 2 shows. What Fig 2 shows is not the slope of the curve, but the mass loss over a certain time interval (so the slope **between** t0 and t30, and not the slope **at** t30).

*Authors : We agree with the comment. But, we computed in the new manuscript new indicators based on the proposal done in comment linked to L337-340. We computed 2 new kinds of indicators: local slope (slope between max and max+30sec, slope between max+30 and max+60,..) and the "delta - t" (ex delta-t50-75, time between 50% loss and 75% weight loss). After analyses, we proposed to switch to the new version of "local slopes", which are relevant and show interesting results. When presenting these new slopes, we will be as clear as possible to avoid misunderstanding.*

- L240 what about testing (Wmax-Wend) ?

*Authors : As we consider the relative soil mass (Wmax = 1), Wmax-Wend won't provide more information than Wend itself.*

- L240 indicators are presented as 'mass at the end' and 'area under the curve'. So units should be 'g' and 'g.sec'. But in later graphs, Wend is presented as unitless and AUC is in 1/sec. So it seems that both indicators have been normalized. This must be explained more clearly.

*Authors : As explained above (see l. 228-229 of the initial version of the text), soil mass is a "relative soil mass", without unit. The 'y-axis' has no unit (relative soil mass, normalized according to Wmax [g/g] = [-]) --> Wend and all other "weight" indicators (Wmax, Wt0,...) have no unit.*

*As a consequence the unit of slopes are only [1 / sec] = [sec$^{-1}$]  and AUC [1 x sec] = [sec].*

*It was expressed in L228-229, but we will present that more clearly (see earlier proposition of reformulation).*

- L241 on line 219, it says the experiment is run for **approximately** 1000 sec. The AUC is evidently going the depend on the length of the experiment. So is a fixed duration used for all samples ?

*Authors: Thank you for this comment. for the sake of comparability, we fixed the time to 900s for each sample, because a few of them didn't last until 1000 s. This 900 s timestep is our reference time for AUC calculation. For QST that had a duration time < 900s because of reaching a steady state before that, the curves were artificially extended to 900 for a comparable AUC between all experiences.*

- The curves are normalized by Wmax, but not by Wend (use Wmax as upper limit and Wend as lower limit for normalization): isn't that introducing some sort of bias ? Also, in the AUC, a large part of the value may come from Wend * duration (the bottom, rectangular area in Fig 2). So a large chunk of the AUC contains the same info as Wend. Normalizing the curves by Wmax and Wend would allow to have an AUC that is independent of Wend.

*Authors:*

*As suggested, the redundancy analysis of QST indicators revealed a strong (r=0.98) positive correlation between AUC and Wend. We made a try to split the AUC into two (the rectangular area delimited by Wend and the area between the curve and Wend. Nevertheless, the curve-dependent fraction of the AUC correlates poorly to MWDs and soil properties. As the interpretation of this indicator is not clear to us, we propose not to use it.*

*We agree that the final time of measurement is very important for the calculation of the AUC because the longer the time considered, the more the rectangular area controls the AUC... At the moment we didn't look to optimize the time considered for AUC calculation with any objective criterion (such as, e.g., sensitivity to soil management practices...).*

*Calculation of the AUC between Wmax and Wend would provide a completely different information from the current AUC. Current AUC decreases with total relative mass loss by disaggregation; the proposed AUC (calculated between Wmax and Wend) would generally increase with mass loss but would also depend on the kinetics of mass stabilization. It would also be somehow redundant with Wend.*

*The contribution of the rectangular area delimited by Wend is important for samples with a good structural stability (elevated Wend) but decreases with soil structural instability (low Wend).*

- L243 check this sentence

*Authors: Sorry for this mistake, sentence will be removed.*

- Caption of Figure 1 : What does 'managing QST laboratory' mean ? Or do you mean 'for managing and graphically displaying QST laboratory data '?

*Authors: The `slaker` application is used also during the QST, for*

- *adding sample identification and useful metadata*
- *Defining experience parameters (max time, …)*
- *starting the recording of data,*
- *Checking the good processing of the data collection*
- *Stopping the experience (computer / data part)*
   o *We agree the sentence is not clear, we propose to adapt : "application for parametrizing and driving the experience"*
- L245 why only the roots remaining in the cage ? Isn't this a biased estimate, as smaller root fragments may have fallen through the mess during sample breakdown ? Wouldn't it be more appropriate to pass the entire soil again through a 2-mm mesh sieve to recover all roots?

*Authors: The idea raised during the experience, seeing roots in the basket. We think you are right and, in further work, we will adapt the protocol for a more rigorous quantification of root content. We certainly underevaluated the root biomass, in the 2 tested treatments of the Tillage LTE. We propose to move the root biomass figure to supporting information.*

- L256 Although the choice of threshold is to some extent arbitrary, using a 10% probability threshold is somewhat unusual in statistics. Using a higher threshold can sometimes be justified based on the dataset characteristics, but what is the reason for this choice here ? Please justify.

*Authors: We will adapt the tables and the text considering only the significatively differences between treatments at a 5% probability threshold, more usual in statistical analyses*

- L268 there is no such Fig. 2. Fig. 2 corresponds to the QST and Fig. 3 to another correlation. Perhaps you mean Table 2?

*Authors: Yes, we should have written Table 2. We will correct the wrong reference in the revised version, the mistake occurs 8 times in the text*

- 3.1 it is also worthwhile noting that correlations are usually stronger with MWD1 than MWD2 (which seems to make sense)

*Authors: This is generally true, except for t75 to t95 which also makes sense, as the end of the curve is not expected to relate to slaking but rather to differential swelling and physico-chemical dispersion.*

- Table 2 : It seems that slope values are expressed in negative values, which was not immediately obvious to me after reading the materials and method section. (I was expecting a higher slope to be negatively correlated with SOC content; the positive correlation stems from the fact that negative values are used). This should be clarified.

*Authors: Slopes (max-30, …) and also new proposed "local slopes" (slope-30-60, …) are all evaluated in the decreasing phase of the QST (with Wmax as the starting point), so with negative values.*

*We will be more explicit int the materials and methods section: we propose to add information at l. 236: "...taking tmax as the starting point (...). The steepest slopes have therefore the most negative values."*

*Higher values of SOC are associated with slope closer to 0. This is illustrated in the following graph given as an example*

[Figure]

o

- L287 : the study of Johannes et al., 2017, did not investigate soil structural stability but soil structural quality (which are two different concepts)! Visual assessment methods don't allow to assess stability.

*Authors : We agree with the precision, we will correct the paragraph in that way:*

*stability changed to quality*

- In section 3.2, I'm missing information regarding the correlation among indicators from the same test. How are slopes, txx, AUC, … correlated in the QST test ? Is it worth considering all these indicators? Same for MWDs and MAs in Le Bissonnais tests : are they correlated ? This comment also relates to L301-304 : what was the basis for this selection of criteria? Correlation analysis would help justify the selection.

*Authors : Thank you for this comment. Indeed we need a clearer decision rule to select indicators from curves to compare the treatments of the trials. After calculation of the new indicators suggested by Reviewer 2 (delta slopes, delta t to be more specific to a certain part of the curve than initial slopes and t), we calculated the correlation matrix for QST indicators (below) and propose to add it as a supporting information.*

*From this matrix we observe that several indicators are highly redundant (r > 0.9):*

- *Wend, AUC Slope 300 & Slope 600*
- *Slopes 30 & Slope 60*
- *Tmax, t25 & t50*
- *T50 & t75, t75 & t90, t90 & t 95*
- *Delta t & t...*

| | Slope 0-max | tmax | Wmax-Wt0 | Slope max-30 | Slope 30-60 | Slope 60-30 | Slope 300-600 | Slope max-60 | Slope max-300 | Slope max-600 | dt max-25 | dt 25-50 | dt 50-75 | dt 75-90 | t50 | t75 | t90 | t95 | Wend | AUC |
|---|---|---|---|---|---|---|---|---|---|---|---|---|---|---|---|---|---|---|---|---|
| Slope 0-max | 1.000 | -0.574 | -0.361 | -0.514 | -0.357 | 0.063 | 0.354 | -0.523 | -0.417 | -0.353 | -0.595 | -0.491 | -0.376 | -0.335 | -0.571 | -0.485 | -0.469 | -0.372 | -0.355 | -0.421 |
| tmax | -0.574 | 1.000 | 0.877 | 0.780 | 0.640 | 0.043 | -0.457 | 0.837 | 0.740 | 0.652 | 0.913 | 0.823 | 0.712 | 0.518 | 0.910 | 0.842 | 0.784 | 0.712 | 0.640 | 0.747 |
| Wmax-Wt0 | -0.361 | 0.877 | 1.000 | 0.711 | 0.478 | 0.089 | -0.273 | 0.718 | 0.657 | 0.615 | 0.724 | 0.537 | 0.494 | 0.489 | 0.665 | 0.599 | 0.617 | 0.609 | 0.584 | 0.669 |
| Slope max-30 | -0.514 | 0.780 | 0.711 | 1.000 | 0.498 | -0.160 | -0.420 | 0.938 | 0.714 | 0.652 | 0.818 | 0.661 | 0.520 | 0.403 | 0.778 | 0.665 | 0.616 | 0.536 | 0.621 | 0.739 |
| Slope 30-60 | -0.357 | 0.640 | 0.478 | 0.498 | 1.000 | 0.271 | -0.315 | 0.764 | 0.814 | 0.746 | 0.564 | 0.618 | 0.501 | 0.221 | 0.614 | 0.581 | 0.474 | 0.348 | 0.772 | 0.809 |
| Slope 60-30 | 0.063 | 0.043 | 0.089 | -0.160 | 0.271 | 1.000 | 0.460 | -0.022 | 0.511 | 0.599 | -0.247 | -0.069 | -0.055 | -0.078 | -0.172 | -0.110 | -0.108 | -0.149 | 0.614 | 0.481 |
| Slope 300-600 | 0.354 | -0.457 | -0.273 | -0.420 | -0.315 | 0.460 | 1.000 | -0.449 | -0.159 | 0.045 | -0.703 | -0.497 | -0.448 | -0.392 | -0.635 | -0.557 | -0.543 | -0.556 | 0.023 | -0.148 |
| Slope max-60 | -0.523 | 0.837 | 0.718 | 0.938 | 0.764 | -0.022 | -0.449 | 1.000 | 0.846 | 0.769 | 0.839 | 0.742 | 0.591 | 0.384 | 0.829 | 0.732 | 0.649 | 0.537 | 0.759 | 0.863 |
| Slope max-300 | -0.417 | 0.740 | 0.657 | 0.714 | 0.814 | 0.511 | -0.159 | 0.846 | 1.000 | 0.975 | 0.594 | 0.602 | 0.477 | 0.292 | 0.623 | 0.570 | 0.502 | 0.387 | 0.978 | 0.998 |
| Slope max-600 | -0.353 | 0.652 | 0.615 | 0.652 | 0.746 | 0.599 | 0.045 | 0.769 | 0.975 | 1.000 | 0.456 | 0.505 | 0.393 | 0.224 | 0.499 | 0.463 | 0.402 | 0.290 | 0.988 | 0.976 |
| dt max-25 | -0.595 | 0.913 | 0.724 | 0.818 | 0.564 | -0.247 | -0.703 | 0.839 | 0.594 | 0.456 | 1.000 | 0.837 | 0.665 | 0.479 | 0.965 | 0.837 | 0.762 | 0.693 | 0.457 | 0.605 |
| dt 25-50 | -0.491 | 0.823 | 0.537 | 0.661 | 0.618 | -0.069 | -0.497 | 0.742 | 0.602 | 0.505 | 0.837 | 1.000 | 0.849 | 0.547 | 0.951 | 0.944 | 0.863 | 0.764 | 0.487 | 0.602 |
| dt 50-75 | -0.376 | 0.712 | 0.494 | 0.520 | 0.501 | -0.055 | -0.448 | 0.591 | 0.477 | 0.393 | 0.665 | 0.849 | 1.000 | 0.574 | 0.782 | 0.961 | 0.886 | 0.869 | 0.344 | 0.471 |
| dt 75-90 | -0.335 | 0.518 | 0.489 | 0.403 | 0.221 | -0.078 | -0.392 | 0.384 | 0.292 | 0.224 | 0.479 | 0.547 | 0.574 | 1.000 | 0.532 | 0.588 | 0.860 | 0.829 | 0.165 | 0.281 |
| t50 | -0.571 | 0.910 | 0.665 | 0.778 | 0.614 | -0.172 | -0.635 | 0.829 | 0.623 | 0.499 | 0.965 | 0.951 | 0.782 | 0.532 | 1.000 | 0.924 | 0.844 | 0.757 | 0.491 | 0.630 |
| t75 | -0.485 | 0.842 | 0.599 | 0.665 | 0.581 | -0.110 | -0.557 | 0.732 | 0.570 | 0.463 | 0.837 | 0.944 | 0.961 | 0.588 | 0.924 | 1.000 | 0.919 | 0.869 | 0.430 | 0.569 |
| t90 | -0.469 | 0.784 | 0.617 | 0.616 | 0.474 | -0.108 | -0.543 | 0.649 | 0.502 | 0.402 | 0.762 | 0.863 | 0.886 | 0.860 | 0.844 | 0.919 | 1.000 | 0.954 | 0.352 | 0.497 |
| t95 | -0.372 | 0.712 | 0.609 | 0.536 | 0.348 | -0.149 | -0.556 | 0.537 | 0.387 | 0.290 | 0.693 | 0.764 | 0.869 | 0.829 | 0.757 | 0.869 | 0.954 | 1.000 | 0.226 | 0.379 |
| Wend | -0.355 | 0.640 | 0.584 | 0.621 | 0.772 | 0.614 | 0.023 | 0.759 | 0.978 | 0.988 | 0.457 | 0.487 | 0.344 | 0.165 | 0.491 | 0.430 | 0.352 | 0.226 | 1.000 | 0.979 |
| AUC | -0.421 | 0.747 | 0.669 | 0.739 | 0.809 | 0.481 | -0.148 | 0.863 | 0.998 | 0.976 | 0.605 | 0.602 | 0.471 | 0.281 | 0.630 | 0.569 | 0.497 | 0.379 | 0.979 | 1.000 |

We propose to select four indicators to compare the different treatments of the three trials based on the following criteria:

- _Avoid highly redundant indicators (r < 0.7), and if arbitration is necessary, choose the conceptually simplest one_
- _Having one indicator of each type/part of the curve_ :
    o _Start of the curve_
    o _Local slopes (e.g. slope 30-60) for the early to intermediate mass loss_
    o _Delta t (e.g. t50-t75) for the later mass loss_
    o _One "global" indicator_
- _Select the most discriminant indicators between treatments_.

According to this decision rules, we propose to keep:

- **Tmax**
- **Slope 30-60** (for the tillage & PK trial) and **slope 60-300** for the SOM trial
- **t50-t75**
- **Wend**

A more exhaustive selection but with already some arbitration (delta slopes & delta t) will be kept for comparison to MWDs & soil properties in Table 2.

We will adapt the result section accordingly.

- Fig. 4 Wmax – Wt0 are given as unitless, even though in materials and methods the authors talk about mass loss. This is confusing. The same problem arise for slopes, Wend, AUC. The confusion seems to stem from the fact that the indicators are not well explained in the M&M section (one speaks of 'mass loss' whereas it should be 'relative mass loss').

*Authors : Indeed, as clarified earlier, all mass in our graphs and indicators are "relative" to the maximum mass (Wmax) reached after immersion. We will clarify in MM section and be more explicit when naming the indicators linked to max.*

  o *So our indicator Wmax – Wt0 is indeed unitless*
  o *More details in the answer of comment here above (l.225, l.240)*

- L336-337 : as mentioned above, it is not correct to attribute single mechanisms to each of the three tests of Le Bissonnais. In the FW-fast wetting test (MWD1), all mechanisms are involved (slaking, dispersion, diff. swelling) except mechanical breakdown. In the SW-slow wetting test (MWD2), dispersion and diff. swelling can be present. So the fact that the QST indicators of L335 are better correlated with FW than SW does not necessarily indicate that slaking is the **main** mechanisms, but merely that slaking significantly contributes to breakdown in the QST, making the correlation stronger with MWD1. This is a subtle but important difference.

*Authors : Thank you for this point. We propose to reformulate accordingly l. 336 : "... , which indicates that slaking significantly contributes to the initial stage of the QST."*

- L337-340 : the 'problem' with t50, t75, etc… is that these indicators inherently also include what happens during the early times. Wouldn't it be much more relevant to consider the time it takes for the sample to loose 25% of its mass, then the time from 25 to 50%, then the time from 50 to 75, etc … ? In this way, the various indicators can be expected to be much more independent, and interpretation of processes would also be facilitated. If indeed dispersion and diff. swelling are the main mechanisms at later stages, then one would expect even stronger correlations between MWD2 and, say, the time needed to lose 75-90% of the mass than between MWD2 and t90.

*Authors : We welcome this remark and the idea of new indicators more independent from each other. We integrated these new indicators in our code and analysed the results. It provided new insights, and we prefer to use them rather than former Slopes & t's that were removed, to avoid highly redundant indicators (see earlier comment on redundancy analysis). We have to rewrite the result section accordingly in the revised version. (see also our answer to comment L234-236).*

- L343 : indeed ! … and from differential swelling. If the authors are aware of this, then why present the Le Bissonnais tests in the wrong way in materials and methods ?

*Authors : Agreed, see above the proposed changes*

- L345 I think the authors misunderstand the FW test of Le Bissonnais. Slaking plays an important role in this test, but it is not the only mechanism at play. It is the difference between the SW and FW tests that predominantly reflects the effect of slaking (though slaking may actually also facilitate dispersion, so the effects are not simply additive). One cannot say that the FW test alone reflects the sole effect of slaking.

*Authors : We take note of these important remarks. We propose the following reformulation l. 343 – 349 : "... from the fast and slow wetting tests of Le Bissonnais. It is also worth to mention that time of wetting of the soils of our study was relatively short (< 2 min), as indicated by the release of air bubbles from soil. We therefore advocate that indicators from the initial stage of the curve provide information much more specific to slaking than the fast wetting test of Le Bissonnais, lasting 10 minutes, which largely exceeds the time during which slaking occurs."*

- L348 indeed, the continuous measurements of QST would allow to better discriminate between mechanism on a single sample, something that cannot be done with the le Bissonnais test. Nevertheless, as suggested above, using time intervals (t0-25, t25-50,t50-75, …) should allow to better discriminate between processes than t0-25, t0-50, t0-75, etc. Furthermore, the difference between MWD1 and MWD2 of Le Bissonnais should better reflect the sole effect of slaking than MWD1 alone, so testing correlations between QST results (early times) and (MWD2-MWD1) might be of interest.

*Authors : Thank you very much for these remarks providing new insights and perspectives of improvement. As answered here above, we computed now all the new proposed indicators and analysed them and we will integrate them in the text. We also explored the difference between MWD1 and MWD2. This will be commented in the new version of the manuscript.*

- The same reasoning would be applied to the slopes : slope max-60, 60-300, 300-600.

*Authors : We fully agree, deltas for slopes were calculated and will be analysed, too*

- L385 : again, it is not correct to state that VESS assesses structural stability (it is a measures of quality)

*Authors : "stability" changed to "quality"*

- L421 : this is rather speculative. It is not certain at all that this would be the case. FYM is very different from green manures or crop residues (much lower C/N ratio right from the start) and therefore will not interact with microbial life in the same way.

*Authors : We agree to remove this speculative statement about the timing of FYM application.*

- L424-425 : I think I sort of understand what the authors mean, but the sentence is awkward and must be clarified

*Authors : Proposition of reformulation l. 423: "...QST indicator (Fig. 6). However, indicators from the late part of QST curves (slope max 300 and slope max 600) and global indicators (Wend, AUC) tend to discriminate better between tillage treatments."*

- L438-440:   Again, I'm not aware that chloride can have such effects on soils with permanent charge.  There is no mention of the effect of chloride on stability in the paper of Paradelo.  It is related to the monovalent cation (K).  This discussion must be revised.

*Authors : Proposition of revision l. 437 ff : "... the working assumption that KCl application might decrease soil structural stability (Paradelo et al. 2016) was not verified. This might be due to a relatively short-lived destructuring effect of KCl, since the last application occurred in the summer of 2016, almost three years before soil sampling. The beneficial effect of K fertilization on crop production and restitution of organic matter to soil might also have counteracted a potentially negative short-term effect."*

- L450 aggregate stability tests such as Le Bissonnais or drop impact tests (Imeson and Vis) where developed in view of relating the results to soil erodibility or even as a way of parameterizing soil detachment during rainfall in erosion models.  In that respect, working with aggregates seems fairly relevant (particularly in case of cultivated tilled soils).  As a means to assess soil quality, the larger sample size used in the QST seems relevant.  Whether QST results could be used to assess soil erodibility remains to be proven, even though the good correlations with Le Bissonnais results are encouraging.

*Authors : Again, we agree with the view of the reviewer. The link between QST and soil erodibility need new experiments/data. We propose to add the following complement of information l. 452:*

*"...inferior or superior equivalent diameter). Nevertheless, the relevance of QST curves to assess soil erodibility needs to be verified."*

- L460 : differential swelling ?

*Authors : proposition of adaptation: "... three main mechanisms … (slaking, physico-chemical dispersion and differential swelling)"*

- L461 see suggestions above on how to possibly improve the indicators

*Authors : We computed already and we will adapt the manuscript accordingly*

- L463-465 : again, there is a bit of a mixup here between 'stability' and 'quality'

*Authors : "stability" changed to "quality" in the revised version*

- L468 : indeed, some overlap is expected with the current indicators, but this could possibly be improved (see suggestions above)

*Authors : Same answer as above, we agree and we will adapt this statement in the revised version*

- L484 I may be wrong (those documents are not easily accessible), but I believe that the equation developed in Laon does not relate to detachment by drop impact per se (but rather to the aggregate stability tests of Henin and Monnier as a measure of the soil sensitivity to crusting). To be checked.

*Authors : Unfortunately, this soil sealing/crusting index is calculated routinely in the services of soil analysis for farmers in Belgium and we know the equation that is applied but we were unable to access the original article, so we were unable to check this point. We therefore propose the following modification l. 482:*

*" However, soil resistance to sealing and crusting is routinely estimated by pedotransfer functions relating using pH in water, SOC content and clay content as input variables (Remy & Marin-Laflèche 1974), which appears complementary with the information offered by the QST."*

- L494-497 : future developments are not to be mentioned in the conclusion

*Authors : Accepted, we will remove these lines in the revised version*

- L503 : I do not believe this was really demonstrated in this paper (as a matter of fact, linear correlations – as demonstrated in this paper – are not compatible with thresholds) , and it should therefore not appear here.

*Authors : we propose to remove the following information from the text l. 503-504:"...with the threshold value of 0.1 being a reasonable target for SOM management at field and farm scale..."*

- See additional annotations in the attached pdf file

*Authors : We thank you very much for this, we will take the annotations into account while preparing our revised manuscript for the next submission.*

**Citation**: https://doi.org/10.5194/egusphere-2022-1092-RC2

---

## Author Response (AR2)

**egusphere-2022-1092 (SOIL) - Answers to reviewers comments (iteration: minor revision)**

F. Vanwindekens, B. Hardy

Sept., 2023

—

**Title** The QuantiSlakeTest, measuring soil structural stability by dynamic underwater weighing

**Author(s)** Frédéric M. Vanwindekens and Brieuc F. Hardy

**MS type** Original research article

**Iteration** Minor revision

—

Dear Reviewer, Dear Editors,

We would like to thank you agin about the time spent for this second revision of our manuscript. Most comments appear to be relevant and will help to strengthen the manuscript.

**Revision 2 - Reviewer comment**

**Suggestions for revision or reasons for rejection**

- Overall, the articles has been much improved in terms of clarity and the authors should be commended for the revisions. There are nevertheless a few minor issues left to address.

    – –> **Thanks a lot for this comment.**

**Tittle**

- "The QuantiSlakeTest, measuring soil structural stability by dynamic underwater weighing" : strictly speaking, the weighing is not done underwater. The sample is underwater. What about "The QuantiSlakeTest, measuring soil structural stability by dynamic weighing of undisturbed samples immersed in water"?

  - –> **We agree with the comment and adopt the proposed title for the article : The QuantiSlakeTest, measuring soil structural stability by dynamic weighing of undisturbed samples immersed in water**

**Abstract**

OK

- –> **Thank you**

**Introduction**

I greatly appreciated the changes brought by the authors. The reading is now more fluid, and the text more focused. Nevertheless, there are some mostly minor issues that need to be addressed. See also the annotated manuscript for typos and corrections of English language.

- –> **Thanks a lot for the general comment. We include the correction of English language suggested and other remarks done in the pdf**

- L24 : please add a reference

  - –> **We added (Lal, 1991)**

- L25-26 : remove 'of Belgium', because this is equally true across all of the western European loess belt; add a reference supporting the fact that structure is particularly relevant for Luvisols (I don't think it is true, but what the authors probably mean is that structure of Luvisols is particularly sensitive to management (due to their texture) and therefore even more attention has to be paid to structure for these soils)

  - –> **We removed Belgium**

- L34 'structural stability' would probably be more relevant, as 'aggregate stability' does not relate to compaction.

  - −> **We changed.**

- L44 : I didn't check the publication of Meersmans et al. 2011, but I don't think these authors are actually at the source of this threshold value. It is good practice to cite original papers rather than citing authors which cited other authors. Though I've also read about the 1.2% SOC threshold for luvisols, a recent paper has reported that this threshold may be closer to 2% SOC (Shi et al., 2020; `https://www.sciencedirect.com/science/article/pii/S0016706119310298`)

  - −> **Thank you for your advise on avoiding bypass when citing scientific articles. The Shi et al paper relies only on the fast wetting test of Le Bissonnais which explains why the limit is higher. The 1.2 or 1.15 value generally considered as a suitable threshold in Belgium comes from a technical report of Van Camp et al. (2004) that is generally considered for loessic Luvisols of Belgium, so we will stick to it.**

- L51-53 : rephrase sentence, because climate and soils are all part of environmental factors (not just topography), while 'soil cover' is more of a management factor (and not an environmental factor).

  - −> **The rephrased sentence is: "Soil erosion is governed by rainfall erosivity, topographic factors, soil cover and intrinsic soil erodibility, depending on several soil properties such as hydraulic conductivity and aggregate stability"**

- L59 significant presence of Al only in soils with pH < 5 (strongly acidic). Even though base saturation will progressively decrease with decreasing pH, Ca and Mg will thus remain important in weakly acidic soils

  - −> **Precision is now given : Al in strongly acidic soils and Ca and Mg in slightly acidic to slightly basic soils.**

- L75 : sounds intuitive, but has this been studied ? A reference would be nice. Compaction most strongly affects macroporosity, which is typically not the porosity found inside small aggregates. Macroporosity

corresponds more to pores in-between aggregates + macropores resulting from biological activity (roots, worms, etc.)

- – –> **As this input had been suggested by the reviewer himself in the first round of review, we were supposing that it was a known cause to effect relationship. We didn't identify literature making this link in our bibliography so we propose to rephrase: "The resistance of soil to mechanical breakdown possibly also improves resistance to soil compaction due to traffic on the field"**

- L75 "differential swelling occurs under wet conditions" : actually, it happens during the process of wetting. In the presence of swelling clays, as soon as free water is added, swelling will occur as a result of water moving in-between the clay platelets.

  - – –> **Rephrased sentence: ". . . occurs during soil wetting"**

- I'm surprised by the statement that differential swelling mainly plays a role at macroscopic scales. I could not find this in the paper of Le Bissonnais 1996 (which is cited here by the authors). On the contrary, it says in that paper To my understanding, differential swelling is a microscopic mechanism (separation of clay platelet) inside clay domains. Because the orientation of the platelets of different domains is more or less random, the swelling generates mechanical stresses inside the macroscopic aggregates.

  - – –> **Rephrased sentence: "Differential swelling plays a role at both macroscopic and microscopic scale and may split the soil into macro- or micro-aggregates."**

- L82-89 : I would have expected a few words about the 'scale' of the units being subjected to aggregate stability measurements : from aggregates of a few mm to . . . (cores) ?

  - – –> **The sentence was completed accordingly : "Traditional methods are destructive and rely on the resistance of soil aggregates to soil undisturbed cores to fragmentation under wet or, less often, dry conditions."**

- L98 : what do you mean by 'delay' ? 'labour requirement' ?

  - – –> **Yes, we adopted the proposed term**

- L100 : throughout the introduction, 'aggregate stability' has been used (almost) exclusively. Here, the authors switch to 'structural stability'. As mentioned in the previous review, I believe this is an adequate use of terms, but for the reader this switch happens insidiously. It would be good to explicitly express this conscious change in terminology. This is also related to the 'scale' issue raised in relation to L82-89.

  - $->$ **We propose one complementary sentence to clarify this point: "The test works on soil cores of a large volume (100 cm$^3$), therefore we consider that QST provides measurements of soil "structural stability" rather than aggregate stability (related to the properties of isolated aggregates)."**

- L109-113 : these lines should be moved to the beginning of the M&M section.

  - $->$ **We moved them**

**Materials and methods**

- L127 I believe plots must always be aligned in rows in a Latin square design

  - $->$ **We removed this information**

- L141 : please specify whether the means are followed by standard deviation or standard error

  - $->$ **we added this info : standard deviation**

- L158 'repeated three times in ... two blocks' ? 'three blocks', I suppose? Please check or explain (I assume 1 block = 1 rep)

  - $->$ **In fact, rach of the two blocks contains 2 repetition. We will be more clear, writing « repeated six times (i.e. three repetitions in each block) ». The total is well 54 plots (=3*2*9) is already stated in the manuscript.**

- L177- L178 : It was initially not clear to me that these oven dried samples were used ONLY for bulk density determination.

  - $->$ **OK, FYI we already tested these samples, but the results is beyond the scope of our proposed manuscript.**

- L212 change to 'namely slaking, mechanical breakdown and differential swelling – clay dispersion . . . '

  - –> **We adapted accordingly**

- L216-219 : this should be moved to the introduction (see earlier comment regarding aggregate / structural stability)

  - –> **The paragraph can be removed since the information is now given previously in the introduction**

- L223 'timestep' (units = time), and not 'frequency' (units = 1/time)

  - –> **OK**

- L245-246 : the nota bene is unclear to me. Please rephrase

  - –> **This is of secondary importance, we removed the nota bene**

- L249 : 900 sec. sounds arbitrary, but probably based on the authors experience of when the mass loss becomes negligible for a majority of samples. Please justify briefly the origin of these '900 sec'.

  - –> **Yes, we added this precision : « These two indicators were calculated for a reference time of (900 s), for which the loss of soil from the basket was negligible for a majority of samples »**

- L251 how were roots separated from soil ? by sieving ? on what mesh size?

  - –> **We simply removed roots retained the « qst basket ». So it is just a global information. We added : « root biomass retained in the 8 mm-mesh metallic basket »**

- L259-260 and L265-266 : the authors explain on L259-260 that they used a linear mixed model approach to evaluate whether soil management affects QST indicators, and on L265-266, they say they used ANOVA for the same purpose. Please clarify.

  - –> **Our text was indeed confusing, sorry for that. We used mixed models (with restricted maximum likelihood as the estimation method), not classical ANOVA (with**

least squares as the estimation method). The results of our F-wald tests are presented in what is still often called an "Anova table", and the function of the package we used is called "Anova", which can indeed cause some confusion. We have removed references to the "Anova" table to avoid ambiguity. The proposed text is now :

In order to test whether soil management practices affect QST indicators, Linear Mixed-Effects Models were fitted. For each model, the QST indicator was used as the outcome variable and the treatments of the trials were used as a fixed explanatory variable, whereas the blocks were defined as a random effect. As several samples were related to one single plot (157 QST in total from 35 plots), the plot identifier was added as a random effect of the model to take into account the dependence between field replicates from one plot.

The normality and the homoscedasticity of the residuals of the models were verified using respectively Shapiro-Wilk and Bartlett tests. For all models, the significance of differences in QST indicators between soil management practices were tested using Type II Wald F tests with Kenward-Roger estimation of degree of freedom (Rcar). When the F-test was significant ($p<0.05$), post-hoc comparisons were performed: treatments of the trial were compared pairwise at (0.05) probability level of significance using estimated marginal means (Remmeans).

All statistical analyses were performed using R version 4.3.0 (2023-04-21) software (R2023). The linear mixed-effect models were performed with the lme4 package (Rlme4), the Wald F tests with the car package (Rcar) and contrast analyses with the emmeans package (Remmeans).

**Results and discusion**

- L291: besides slope 0-max, slope 60-300 and slope 300-600 are also NOT positively correlated to MWD1 according to the table.

  - −> We corrected : « Except for Slope~0-max~, Slope~60-300~ and Slope~300-600~, a positive correlation was found between »

- L324 : how could slope affect structural stability (at plot scale) ?

– −> **The sentence was adapted: "...soil conditions of the sampling sites (namely, microsite heterogeneity related e.g. to microrelief, the presence of crop residues, roots, earthworm galleries, ...)"**

- L330-333 : this sentence is not very clear (':' is used twice in the same sentence), and seems to repeat what has been said on L288-289. Repeating the info is useful, but the sentence can be simplified (see annotated manuscript)

  – −> **We remode the last part of the sentence, like proposed by the reviewer in the pdf.**

Fig. 5 : please indicate on the graphs the results of the pairwise comparisons between treatments (using for instance small cap letters) to better highlight which treatment is different from which other treatment.

- −> **We did it**

- L340 : 'discordant' doesn't sound right in English, but can't think of the proper word. 'Antagonistic results' ?

  – −> **We adopted the proposed word**

- L349 : please express root biomass in terms of root mass density (mg/cm$^3$ or g/dm$^3$ or ...) for easier comparison with the literature

  – −> **We expressed it in the new version of the manuscript**

- L350 name of the variable is misspelled

  – −> **We corrected**

Isn't Fig. 6b the same as Fig. 7d ? I suggest removing 6b.

- −> **Yes the figures are based on the same data, but they have different objectives : 7 is results. 6b is for the sake of pedagogy, illustrating how we come from curves to boxplot of the Wend indicator. We propose to leave the right part of Fig. 6 (6b)**

In Fig. 7, please indicate on the graphs the results of the pairwise comparisons between treatments

- −> **We did it**

- L372 'advocate' means 'publicly recommend'. This is probably not what the authors had in mind, since the sentence is not a 'recommendation'. Please check

    - —> **We change to « We argue »**

- L392 : what does a 'field gradient' mean ? 'field' is not a property (temperature gradient, concentration gradient, . . . )

    - —> **We changed to « concentration gradient »**

- L402 : I don't understand the statement 'with an average complexation potential of 1 g of SOM for 10 g of clay', especially given the first part of the sentence. The next sentence seems to imply that 1 g SOM for 10 g clay is a threshold value. Please explain and rephrase this sentence.

    - —> **The sentence was rephrased: ". . . who found that, for a variety of soils from France and Poland, about 1 g of SOM was necessary to decrease the dispersive power of 10 g of clay by organo-mineral association." Hope this reads better now. If confusion persists please read Dexter et al. (2008)**

- L417-418 : not sure why the authors advocate the use of the 0.1 SOM/clay threshold value, even though their results do not support the existence of such a threshold, and the results of Johannes et al. (2017) and Prout et al. (2020) also do not support the existence of a "threshold" (but rather a linear relationship, as stated by the authors on L405). It may still be that the 0.1 threshold value coincides with a change in structural quality class according to VESS (class boundaries being often more or less arbitrary), but there is nothing in this paper to support the existence of a threshold.

    - —> **The sentence was rephrased to present the interest of the SOC:Clay ratio in a more general way : "To sum up, we suggest that the SOC:clay ratio, a proxy for soil intrinsic 'potential' structural stability, is a valuable indicator of the organic and structural status of agricultural soils (Dexter2008, Johannes2017, Prout2020)."**

- L455 replace 'advocate' by a more suitable word

- – –> we changed by « higher root density and SOC contents in the topsoil also positively impact the biological activity »

- L459-462 : did the authors check the composition of the exchange complex ? Is the proportion of K significantly (and substantially) higher in K2 than K0 treatment, for instance ? As stated by the authors, it may well be that the exchange complex has had time to reequilibrate since 2016, and looking at the relative importance of Ca, Mg and K should give a clue.

  - – –> **Unfortunately we didn't measure plant available or exchangeable cations and CEC.**

- L473 'crumbled' ? (rather than crambled)

**Yes we changed the term, it was already « crumbled » in other paragraphs of our manuscript. Here it was a typing mistake**

- L475 : sampling stony soils with a kopecki ring is very tricky . . . !

  - – –> **We completed the sentence: ". . . for which the adequacy of the sampling procedure and running of the test needs to be verified"**

- L488 '. . . but curve modelling is another perspective of curve interpretation' : replace by '. . . but curve modeling may offer further perspective for curve interpretation'

  - – –> **We changed**

- L496-497 : summing up before a conclusion seems redundant

  - – –> **We removed the sentence as suggested in the pdf alsoo**

**Conclusions**

Some sentences are almost exact copies of sentences used in the discussion, which should be avoided.

- L511 : 'is closely related to the SOM status of soil, well-captured by the SOC:clay ratio' : sounds a bit contradictory. So it's not just the 'SOM status', but the concentration of SOM relative to the clay content.

– –> The following sentence was rephrased to avoid repetition with the discussion and clarify the point raised about the SOM status: L510 ff. "We found that soil resistance to disaggregation correlates positively with SOM content and negatively to clay content, making the SOC:clay ratio a key indicator of the potential structural stability for the soils of central Belgium."

---

## Author Response (AR3)

**egusphere-2022-1092 (SOIL) - Message (step : Final acceptance)**

F. Vanwindekens, B. Hardy

Sept., 2023

—

**Title** The QuantiSlakeTest, measuring soil structural stability by dynamic weighing of undisturbed samples immersed in water

**Author(s)** Frédéric M. Vanwindekens and Brieuc F. Hardy

**MS type** Original research article

**Iteration** Accepted for final publication in SOIL

—

Dear Reviewer, Dear Editors,
We would like to thank you for announcing the acceptance of our manuscript for final publication in SOIL.
Kind regards,
Frédéric Vanwindekens
Brieuc Hardy